# Distributed Inverse Constrained Reinforcement Learning for Multi-agent Systems

**Shicheng Liu & Minghui Zhu**
School of Electrical Engineering and Computer Science
Pennsylvania State University
University Park, PA 16802, USA
{sfl5539,muz16}@psu.edu

## Abstract

This paper considers the problem of recovering the policies of multiple interacting experts by estimating their reward functions and constraints where the demonstration data of the experts is distributed to a group of learners. We formulate this problem as a distributed bi-level optimization problem and propose a novel bi-level "distributed inverse constrained reinforcement learning" (D-ICRL) algorithm that allows the learners to collaboratively estimate the constraints in the outer loop and learn the corresponding policies and reward functions in the inner loop from the distributed demonstrations through intermittent communications. We formally guarantee that the distributed learners asymptotically achieve consensus which belongs to the set of stationary points of the bi-level optimization problem. Simulations are done to validate the proposed algorithm.

## 1   Introduction

Multi-agent systems (MASs) have become an effective way to model large-scale networked systems, *e.g.*, multi-robot systems and the Internet-of-Things. Multi-agent reinforcement learning (MARL) is a systematic framework to coordinate MASs. It extends RL to address the decision-making problem of multiple interacting agents [1]. One challenge in (MA)RL is that manually crafting the reward functions or predesigning the policies can be infeasible for humans when the tasks are too complicated. Inverse reinforcement learning (IRL) presents a way to tackle this problem where a learner aims to recover a reward function (and a corresponding policy) that best explains the behavior in the demonstrations of an expert. Current state-of-the-arts on IRL [2, 3, 4, 5] can successfully solve the reward function inference problem but still face two other major challenges in MARL and MASs: (i) it is often the case that the expected behavior (experts' behavior) is more precisely explained by a reward function combined with a set of constraints instead of a single reward function; (ii) the demonstration data in MASs is usually distributed over multiple learners and unable to be shared due to some practical reasons including privacy concerns and communication burdens. This paper aims to bridge the gap and proposes the first algorithm to solve the three challenges (i.e., reward function inference, constraint inference, and distributed demonstration data) at once.

**Related works**. IRL is an ambiguous problem [6] as different reward functions could explain the demonstrations. To solve this problem, several fundamental approaches are proposed including feature expectation matching [7], maximum margin planning [8], maximum entropy (ME) [9], and maximum causal entropy (MCE) [10]. Recently, some advanced techniques baring more features are proposed. Paper [2] uses adversarial learning to learn reward functions robust to environmental changes, paper [4] applies strongly convex regularizers to the learner's policy to avoid the ambiguity problem, and paper [5] proposes a scalable Bayesian-based method working for large control tasks.

36th Conference on Neural Information Processing Systems (NeurIPS 2022).

In real-world settings, numerous constraints are present and a single reward function is not enough to explain complicated behavior. Paper [11] extends ME IRL to environments with constraints, formulates the constraint inference problem as a maximum likelihood estimation (MLE) problem, and leverages a greedy method to solve the problem. Paper [12] extends paper [11] to continuous environments. These two works both assume the access to the ground truth rewards.

While the discussions in the last two paragraphs are limited to single agents, there have been recent works which extend IRL techniques to multiple agents [13, 14, 15, 16, 17]. Notice that previous works on multi-agent IRL do not explicitly distinguish the notions of "experts" and "learners", their "multiple agents" means "multiple experts" and they only consider centralized learning (*i.e.*, a single learner who obtains all demonstrated behavior of the multiple experts). In this paper, we explicitly distinguish the notions of "experts" and "learners", and use "multiple experts" to represent the commonly used "multiple agents" in literature. Moreover, the aforementioned papers do not consider the more general situation where the demonstrations are distributed over multiple learners and unable to be shared. To handle this situation, the learners need to perform distributed learning.

As a basic topology in distributed learning, peer-to-peer (P2P) architecture has no central server and each node can communicate with a subset of all the nodes. It reduces the risk of single points of failures [18] in the system. This paper proposes D-ICRL based on the P2P structure where ICRL can learn policies by estimating the reward functions and constraints through demonstrations and distributed learners can perform ICRL without sharing the distributed demonstration data.

In specific, we formulate D-ICRL as a distributed bi-level optimization problem where the learners cooperatively learn the constraints through the outer problem and estimate the corresponding reward functions and policies through the inner problem. Paper [19] provides asymptotic convergence for centralized bi-level optimization problems by assuming that the inner problem can be completely solved and its optimal solution can be obtained. Papers [20, 21, 22] relax this assumption by partially solving the inner problem and provide convergence rate analysis for problems with strongly convex inner objective function. Paper [23] solves a special case of distributed bi-level optimization where the outer and inner problems use identical decision variables. To the best of our knowledge, there is no work on bi-level optimization which (i) assumes that the inner objective function is merely strictly convex and partially solves the inner problem; (ii) solves the general form of bi-level optimization in a distributed way. Notice that these two challenges arise in our D-ICRL problem.

**Contribution statement**. Our contributions are threefold. First, we extend IRL to a multi-expert-multi-learner (MEML) setting where a group of learners collaboratively learn policies by estimating the reward functions and constraints without sharing the distributed demonstrations performed by a set of cooperative experts. We formulate this D-ICRL problem as a distributed bi-level optimization problem. Second, we propose a distributed bi-level learning algorithm to solve this problem where the learners cooperatively estimate the constraints in the outer problem and learn the corresponding reward functions and policies in the inner problem. Third, we provide convergence rate analysis to the optimal solution of the inner problem and asymptotic convergence to the set of stationary points of the outer problem. Simulations are conducted to validate the proposed method.

## 2 Model

In this section, we present the models for cooperative experts and distributed learners.

**Experts**. There are $N_E$ experts whose decision making is based on a constrained Markov game (CMG) [24]. A CMG $(\mathcal{S}, \mathcal{A}, \gamma, P_0, P, r_E, c_E, b)$ is defined via state set $\mathcal{S} \triangleq \prod_{i=1}^{N_E} \mathcal{S}^{[i]}$, action set $\mathcal{A} \triangleq \prod_{i=1}^{N_E} \mathcal{A}^{[i]}$, discount factor $\gamma \in (0, 1)$, and initial state distribution $P_0$. The state-action space can be either discrete or continuous. The system state transition function is given by $P$ such that the probability (or probability density in continuous state-action space) of state transition to $s'$ from $s$ by taking action $a \triangleq (a^{[1]}, \cdots, a^{[N_E]})$ is $P(s'|s, a)$. The reward function of expert $i$ is $r_E^{[i]} : \mathcal{S} \times \mathcal{A} \to \mathbb{R}$ and the experts are cooperative (*i.e.*, $r_E \triangleq \sum_{i=1}^{N_E} r_E^{[i]}$). The cost function of expert $i$ is $c_E^{[i]} : \mathcal{S} \times \mathcal{A} \to \mathbb{R}_+$ and the cost function of the whole system is $c_E \triangleq \sum_{i=1}^{N_E} c_E^{[i]}$. Expert $i$'s policy $\pi_E^{[i]}(a^{[i]}|s)$ represents the probability (or probability density) of expert $i$ taking action $a^{[i]}$ at state $s$ and the joint policy of all the experts is denoted by $\pi_E(a|s) \triangleq \prod_{i=1}^{N_E} \pi_E^{[i]}(a^{[i]}|s)$. Define $J_{r_E}(\pi) \triangleq E_{S,A}^{\pi}[\sum_{t=0}^{\infty} \gamma^t r_E(S_t, A_t)]$ as the expected cumulative reward where the initial

state is drawn from $P_0$ and $J_{c_E}(\pi) \triangleq E_{S,A}^\pi[\sum_{t=0}^\infty \gamma^t c_E(S_t, A_t)]$ as the expected cumulative cost. The experts' policy $\pi_E$ wants to maximize $J_{r_E}(\pi)$ subject to $J_{c_E}(\pi) \leq b$. Define the constraint set indicated by $c_E^{[i]}$ as $\mathcal{C}_E^{[i]} \triangleq \{(s,a) \in \mathcal{S} \times \mathcal{A} : c_E^{[i]}(s,a) > 0\}$. Following [12], we study the case where $b = 0$, i.e., hard constraint. Thus, the constraint reduces to $J_{c_E}(\pi) = 0$, implying that the probability of reaching any $(s,a) \in \bigcup_i \mathcal{C}_E^{[i]}$ is zero under policy $\pi$. These experts use $\pi_E$ to demonstrate a set $\mathcal{D} \triangleq \{\zeta^j\}_{j=1}^m$ of $m$ trajectories, which is partitioned into $N_L$ subsets. Each trajectory $\zeta^j \triangleq s_0^j, a_0^j, s_1^j, a_1^j \cdots$ is a state-action sequence of the experts.

**Learners**. There is a group of $N_L$ learners where each learner $v$ knows $(\mathcal{S}, \mathcal{A}, \gamma, P_0, P, b, \mathcal{D}^{[v]})$ and can communicate with other learners. The local demonstration subset $\mathcal{D}^{[v]}$ has $m^{[v]}$ trajectories. The learners choose an $l_r^{[i]}$-dimensional reward feature vector $\phi_r^{[i]} : \mathcal{S} \times \mathcal{A} \to [0, d_1]^{l_r^{[i]}}$ for expert $i$ where $d_1$ is constant and it is assumed that $r_E^{[i]} = (\omega_{r_E}^{[i]})^\top \phi_r^{[i]}$, where $\omega_{r_E}^{[i]} \in \mathbb{R}^{l_r^{[i]}}$ is the (unknown) reward weight vector. The learners also choose an $l_c^{[i]}$-dimensional cost feature vector $\phi_c^{[i]} : \mathcal{S} \times \mathcal{A} \to [0, d_2]^{l_c^{[i]}}$ where $d_2$ is constant and it is assumed that $c_E^{[i]} = (\omega_{c_E}^{[i]})^\top \phi_c^{[i]}$, where $\omega_{c_E}^{[i]}$ is the (unknown) cost weight vector and $\omega_{c_E,j}^{[i]}$ is the $j$-th component of $\omega_{c_E}^{[i]}$. Let $\mathcal{C}_j^{[i]} \triangleq \{(s,a) \in \mathcal{S} \times \mathcal{A} : \phi_{c,j}^{[i]}(s,a) > 0\}$ be the constraint set indicated by the $j$-th cost feature $\phi_{c,j}^{[i]}$ of expert $i$. As the budget $b = 0$, the effect of $\omega_{c_E,j}^{[i]} > 0$ is the same, i.e., $\mathcal{C}_j^{[i]}$ cannot be visited, thus the learners assume $\omega_{c_E,j}^{[i]} \in [0, 1]$.

**Communication network**. The learners execute a two-time-scale learning algorithm over a communication network. The fast-time-scale (inner) algorithm is executed at discrete time $k$ over a time-varying directed graph $\mathcal{G}(k) \triangleq (\mathcal{V}, \mathcal{E}(k))$ where $\mathcal{V} \triangleq \{1, \cdots, N_L\}$ is the node (learner) set and $\mathcal{E}(k) \subseteq \mathcal{V} \times \mathcal{V}$ is the set of directed edges (communication links) at time $k$. The edge $(v, v') \in \mathcal{E}(k)$ means that learner $v$ receives information from learner $v'$ at time $k$ and $(v, v) \in \mathcal{E}(k)$ for all $k \geq 0$. The adjacency matrix of the graph at time $k$ is $W(k) \triangleq [W^{[vv']}(k)]_{v,v' \in \mathcal{V}} \in \mathbb{R}^{N_L \times N_L}$ where $W^{[vv']}(k) = 0$ if and only if $(v, v') \notin \mathcal{E}(k)$. We use $\mathcal{N}^{[v]}(k) \triangleq \{v' \in \mathcal{V} | (v, v') \in \mathcal{E}(k)\}$ to denote the set of neighbors of learner $v$ at time $k$. The slow-time-scale (outer) algorithm is carried out at discrete time $n$ over a graph $\bar{\mathcal{G}}(n) \triangleq (\mathcal{V}, \bar{\mathcal{E}}(n))$, where $\bar{\mathcal{E}}(n)$, $\bar{W}(n)$ and $\bar{\mathcal{N}}^{[v]}(n)$ are defined in an analogous way. The time unit for $n$ is much larger than that for $k$.

**Assumption 1.** *There exists an integer $B \geq 1$ such that the graph $(\mathcal{V}, \mathcal{E}(k) \cup \cdots \cup \mathcal{E}(k + B - 1))$ is strongly connected for all $k \geq 0$.*

**Assumption 2.** *The adjacency matrix $W(k)$ has the following properties:*
*(i) (doubly stochastic) $\mathbf{1}^\top W(k) = \mathbf{1}^\top$ and $W(k)\mathbf{1} = \mathbf{1}$ where $\mathbf{1}$ is the column vector whose entries are all ones. (ii) (non-degenerate) There is an $\epsilon \in (0, 1)$ such that $W^{[vv]}(k) \geq \epsilon$ for all $v \in \mathcal{V}$ and $W^{[vv']}(k) \geq \epsilon$ if $(v, v') \in \mathcal{E}(k)$.*

**Assumption 3.** *The set $\bar{\mathcal{N}}^{[v]}(n) = \mathcal{V}$ for any $v \in \mathcal{V}$ and $n \geq 0$. The adjacency matrix $\bar{W}(n)$ is doubly stochastic and non-degenerate with constant $\bar{\epsilon}$.*

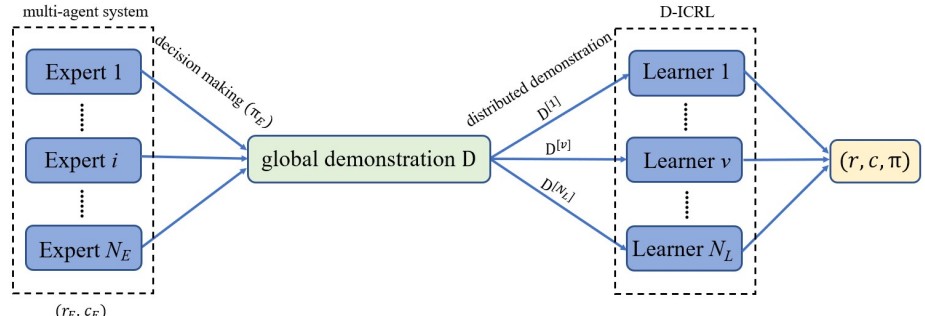

Figure 1: Relation between experts and learners

Figure 1 provides an illustration of the relation between the experts and the learners. The experts operate in a multi-agent system and demonstrate a set $\mathcal{D}$ of trajectories and the learners stand outside the system where each learner $v$ can observe a subset $\mathcal{D}^{[v]}$ of the global demonstration.

# 3 Notions and notations

This section provides notions and notations which will be used in the remaining of the paper.

**Reward feature expectation matching (RFEM)**. Given policy $\pi$, the reward feature expectation vector of expert $i$ is defined as $\mu_r^{[i]}(\pi) \triangleq E_{S,A}^{\pi}[\sum_{t=0}^{\infty} \gamma^t \phi_r^{[i]}(S_t, A_t)]$ and the demonstration $\mathcal{D}$ is needed to estimate $\mu_r^{[i]}(\pi_E)$: $\hat{\mu}_r^{[i]} \triangleq \frac{1}{m} \sum_{j=1}^{m} \sum_{t=0}^{\infty} \gamma^t \phi_r^{[i]}(s_t^j, a_t^j)$, where $(s_t^j, a_t^j) \in \zeta^j$. The RFEM requires $\mu_r^{[i]}(\pi) = \hat{\mu}_r^{[i]}$ for every expert $i$. However, each learner $v$ can only observe the subset $\mathcal{D}^{[v]}$ and its local estimate is $\hat{\mu}_r^{[i,v]} \triangleq \frac{1}{m^{[v]}} \sum_{\zeta^j \in \mathcal{D}^{[v]}} \sum_{t=0}^{\infty} \gamma^t \phi_r^{[i]}(s_t^j, a_t^j)$.

**Cost expectation matching (CEM)**. Similar to RFEM, CEM requires $J_{c_E}(\pi) = J_{c_E}(\pi_E)$. As the true cost function $c_E$ is unknown, given an estimate $c_{\omega_c} = \sum_{i=1}^{N_E} (\omega_c^{[i]})^\top \phi_c^{[i]}$, we define the empirical cost estimate of the experts as $\hat{b}_{\omega_c} \triangleq \frac{1}{m} \sum_{j=1}^{m} \sum_{t=0}^{\infty} \gamma^t c_{\omega_c}(s_t^j, a_t^j)$. The local estimate from learner $v$ is $\hat{b}_{\omega_c}^{[v]} \triangleq \frac{1}{m^{[v]}} \sum_{\zeta^j \in \mathcal{D}^{[v]}} \sum_{t=0}^{\infty} \gamma^t c_{\omega_c}(s_t^j, a_t^j)$.

**Remark 1.** *As RFEM is a standard notion in IRL literature, it is "natural" to follow this and use "cost feature expectation matching" (CFEM). However, here we use CEM instead of CFEM because we want to leverage the budget information provided in CMG. We still use RFEM instead of "reward expectation matching" (REM) because REM does not embed any additional useful information as CMG does not have any "budget information" for cumulative reward.*

**Notations**. We use $\hat{\mu}_r \triangleq [(\hat{\mu}_r^{[1]})^\top, \cdots, (\hat{\mu}_r^{[N_E]})^\top]^\top$ to represent the concatenated empirical reward feature expectation vector. Similarly, we define learner $v$'s concatenated empirical reward feature expectation vector estimate $\hat{\mu}_r^{[v]}$, concatenated reward feature expectation vector $\mu_r(\pi)$ under $\pi$, and concatenated cost feature vector $\phi_c$. Let $\Pi$ be the set of all valid stochastic policies such that every $\pi \in \Pi$ satisfies $\pi(a|s) \geq 0$ for any $s \in \mathcal{S}$ and $a \in \mathcal{A}$ and $\int_{a \in \mathcal{A}} \pi(a|s) da = 1$ for any $s \in \mathcal{S}$. We define the expected cumulative cost of policy $\pi$ under cost function estimate $c_{\omega_c}$ as $J_{\omega_c}(\pi) \triangleq E_{S,A}^{\pi}[\sum_{t=0}^{\infty} \gamma^t c_{\omega_c}(S_t, A_t)]$ and use $\text{Project}_{\Omega_c}(\cdot)$ to denote the projection onto the set $\Omega_c$.

# 4 Problem formulation

In an MEML D-ICRL problem, each learner $v$ aims to use all the available information to collaboratively recover the policy $\pi_E$ through communications by estimating the reward function $r_E$ and the cost function $c_E$. In the learning process, learner $v$ is unwilling to share its private data (*i.e.*, local demonstration and local estimate of reward feature expectation and cost expectation). Notice that estimating $c_E$ is in essence estimating the constraint set $\bigcup_i \mathcal{C}_E^{[i]}$. We formulate the constraint learning problem as a distributed MLE problem:

$$\max_{\omega_c \in \Omega_c} \quad F(\omega_c) = \sum_{v=1}^{N_L} F^{[v]}(\omega_c), \tag{1}$$

where $\Omega_c \triangleq [0,1]^{\sum_{i=1}^{N_E} l_c^{[i]}}$, $F^{[v]}(\omega_c) \triangleq \sum_{j \in \mathcal{D}^{[v]}} \sum_{t=0}^{\infty} \gamma^t \ln \pi_{\omega_c}(a_t^j | s_t^j)$ is learner $v$'s local likelihood function and $\pi_{\omega_c}$ is a policy parameterized by $\omega_c$. To find a statistical distribution model for $\pi_{\omega_c}$, we formulate the following problem based on MCE scheme:

$$\max_{\pi \in \Pi} \quad H(\pi), \qquad \text{s.t.} \quad \mu_r^{[i]}(\pi) = \hat{\mu}_r^{[i]} \quad \forall i \in \mathcal{I}, \quad J_{\omega_c}(\pi) = \hat{b}_{\omega_c}, \tag{2}$$

where $H(\pi) \triangleq \sum_{t=0}^{\infty} E_{S,A}[-\gamma^t \ln \pi(A_t|S_t)]$ is the (infinite-horizon) causal entropy [25].

**Remark 2.** *Directly matching the ground truth budget (i.e., $J_{\omega_c}(\pi) = b$) may cause infeasibility of (2), i.e., the feasible set of (2) is not guaranteed to be nonempty. Therefore, we match $\hat{b}_{\omega_c}$, which is an estimate of $J_{\omega_c}(\pi_E)$, to ensure the feasibility because now, $\pi_E$ is a feasible solution of (2).*

The dual function of (2) is $G(\omega_r, \lambda; \omega_c) \triangleq \max_{\pi \in \Pi} H(\pi) + \sum_{i=1}^{N_E} (\omega_r^{[i]})^\top (\mu_r^{[i]}(\pi) - \hat{\mu}_r^{[i]}) + \lambda(J_{\omega_c}(\pi) - \hat{b}_{\omega_c})$ where the dual variable $\omega_r = [(\omega_r^{[1]})^\top, \cdots, (\omega_r^{[N_E]})^\top]^\top$ is to estimate $\omega_{r_E}$. Let $\eta \triangleq [\omega_r^\top, \lambda]^\top$, it is well-known that $G(\eta; \omega_c)$ is convex in $\eta$ since it is the pointwise maximum of a family of affine functions of $\eta$. As $G(\eta; \omega_c)$ is global, we introduce the local convex dual function $G^{[v]}(\eta; \omega_c) \triangleq \max_{\pi \in \Pi} H(\pi) + \sum_{i=1}^{N_E} (\omega_r^{[i]})^\top (\mu_r^{[i]}(\pi) - \hat{\mu}_r^{[i,v]}) + \lambda(J_{\omega_c}(\pi) - \hat{b}_{\omega_c}^{[v]})$.

**Lemma 1.** *(i) The optimal solution of problem (2) is the constrained soft Bellman policy* $\pi_{\eta^*(\omega_c);\omega_c}$*, where* $\eta^*(\omega_c)$ *is an optimal solution of* $\min_\eta G(\eta;\omega_c)$*. (ii) The local dual function* $G^{[v]}(\eta;\omega_c)$ *is differentiable and its gradient is* $[(\mu_r(\pi_{\eta;\omega_c}) - \hat{\mu}_r^{[v]})^\top, J_{\omega_c}(\pi_{\eta;\omega_c}) - \hat{b}_{\omega_c}^{[v]}]^\top$*.*

The proof of Lemma 1 is similar to [10, 25, 26]. For the sake of completeness, we still include the proof and the expression of the constrained soft Bellman policy in the Appendix.

Lemma 1 indicates that $\pi_{\eta^*(\omega_c);\omega_c}$ is the statistical model we need in problem (1). Therefore, problem (1) can be reformulated as the following distributed bi-level optimization problem:

$$\max_{\omega_c \in \Omega_c} \quad F(\omega_c, \eta^*(\omega_c)) = \sum_{v=1}^{N_L} F^{[v]}(\omega_c, \eta^*(\omega_c)), \tag{3}$$

$$\text{s.t.} \quad \eta^*(\omega_c) = \arg\min_\eta \sum_{v=1}^{N_L} m^{[v]} G^{[v]}(\eta;\omega_c), \tag{4}$$

where $F^{[v]}$ and $m^{[v]} G^{[v]}$ are known only to learner $v$.

**Remark 3.** *Problem (4) and* $\arg\min_\eta G(\eta;\omega_c)$ *are equivalent because* $\sum_{v=1}^{N_L} m^{[v]} G^{[v]}(\eta;\omega_c) = \sum_{v=1}^{N_L} m^{[v]} H(\pi_{\eta;\omega_c}) + \sum_{v=1}^{N_L} m^{[v]} \sum_{i=1}^{N_E} (\omega_r^{[i]})^\top (\mu_r^{[i]}(\pi_{\eta;\omega_c}) - \hat{\mu}_r^{[i,v]}) + \lambda \sum_{v=1}^{N_L} m^{[v]} (J_{\omega_c}(\pi_{\eta;\omega_c}) - \hat{b}_{\omega_c}^{[v]}) = m H(\pi_{\eta;\omega_c}) + m \sum_{i=1}^{N_E} (\omega_r^{[i]})^\top (\mu_r^{[i]}(\pi_{\eta;\omega_c}) - \hat{\mu}_r^{[i]}) + \lambda m (J_{\omega_c}(\pi_{\eta;\omega_c}) - \hat{b}_{\omega_c}) = m G(\eta;\omega_c).$

## 5  Algorithm and convergence guarantee

In this section, we develop a bi-level distributed learning algorithm to solve problem (3)-(4) and provide convergence rate to the optimal solution of the inner problem (4) and asymptotic convergence to the set of stationary points of the outer problem (3).

---

**Algorithm 1** MEML D-ICRL

---

**Input**: $\{\omega_c^{[v]}(0)\}_{v=1}^{N_L}, \{\eta^{[v]}(0)\}_{v=1}^{N_L}, W(k), \bar{W}(n)$

**Output**: $\omega_c^{[v]}(n), \bar{\eta}^{[v]}(\omega_c^{[v]}(n)), \pi_{\bar{\eta}^{[v]}(\omega_c^{[v]}(n));\omega_c^{[v]}(n)}$  $\quad \forall v \in \mathcal{V}$

1: **for** $n = 0, 1, \cdots$ **do**
2:     **for** $v \in \mathcal{V}$ **do**
3:         Receives $\omega_c^{[v']}(n)$ from $v' \in \bar{\mathcal{N}}^{[v]}(n)$
4:         $\bar{\eta}^{[v]}(\omega_c^{[v]}(n)), \pi_{\bar{\eta}^{[v]}(\omega_c^{[v]}(n));\omega_c^{[v]}(n)} = $ Inner process$(\omega_c^{[v]}(n))$
5:         **if** $n = 0$ **then**
6:             $\bar{\nabla}^{[v]}(0) = \bar{\nabla} F^{[v]}(\omega_c^{[v]}(0), \bar{\eta}^{[v]}(\omega_c^{[v]}(0)))$
7:         **else**
8:             Receives $\bar{\nabla}^{[v']}(n-1)$ from $v' \in \bar{\mathcal{N}}^{[v]}(n)$
9:             $\bar{\nabla}^{[v]}(n) = \sum_{v'=1}^{N_L} \bar{W}^{[vv']}(n) \bar{\nabla}^{[v']}(n-1) + \bar{\nabla} F^{[v]}(\omega_c^{[v]}(n), \bar{\eta}^{[v]}(\omega_c^{[v]}(n))) - \bar{\nabla} F^{[v]}(\omega_c^{[v]}(n-1), \bar{\eta}^{[v]}(\omega_c^{[v]}(n-1)))$
10:         **end if**
11:         $\tilde{\omega}_c^{[v]}(n) = \text{Project}_{\Omega_c}\left(\omega_c^{[v]}(n) + N_L \bar{\nabla}^{[v]}(n)\right)$
12:         $\omega_c^{[v]}(n+1/2) = \omega_c^{[v]}(n) + \beta(n)(\tilde{\omega}_c^{[v]}(n) - \omega_c^{[v]}(n))$
13:         Receives $\omega_c^{[v']}(n+1/2)$ from $v' \in \bar{\mathcal{N}}^{[v]}(n)$
14:         $\omega_c^{[v]}(n+1) = \sum_{v'=1}^{N_L} \bar{W}^{[vv']}(n) \omega_c^{[v']}(n+1/2)$
15:     **end for**
16: **end for**

---

At outer iteration $n$ in Algorithm 1, the learners first cooperatively solve the inner problem for each learner through *Inner process* (lines 3-4) and then use the obtained result $\bar{\eta}^{[v]}(\omega_c^{[v]}(n))$ to

collaboratively solve the outer problem (3) through the outer process (lines 5-14). In what follows, we will elaborate each process.

---

**Algorithm 2** Inner process($\omega_c^{[v]}$)

---

**Input**: $\omega_c^{[v]}$, $\{\eta^{[v]}(0)\}_{v=1}^{N_L}$, $W(k)$
**Output**: $\bar{\eta}^{[v]}(\omega_c^{[v]})$, $\pi_{\bar{\eta}^{[v]}(\omega_c^{[v]});\omega_c^{[v]}}$

1: **for** $k = 0, 1, \cdots, K - 2$ **do**
2:     **for** $\bar{v} \in \mathcal{V}$ **do**
3:         Receives $\eta^{[\bar{v}']}(\omega_c^{[v]}, k)$ from $\bar{v}' \in \mathcal{N}^{[\bar{v}]}(k)$
4:         $\eta^{[\bar{v}]}(\omega_c^{[v]}, k+1) = \sum_{v'=1}^{N_L} W^{[\bar{v}\bar{v}']}(k)\eta^{[\bar{v}']}(\omega_c^{[v]}, k) - \alpha(k)m^{[\bar{v}]}\nabla_\eta G^{[\bar{v}]}(\eta^{[\bar{v}]}(\omega_c^{[v]}, k); \omega_c^{[v]})$
5:     **end for**
6: **end for**
7: $\bar{\eta}^{[v]}(\omega_c^{[v]}) \triangleq \frac{\sum_{k=0}^{K-1} \alpha(k)\eta^{[v]}(\omega_c^{[v]}, k)}{\sum_{k=0}^{K-1} \alpha(k)}$
8: $\pi_{\bar{\eta}^{[v]}(\omega_c^{[v]});\omega_c^{[v]}}$ is the corresponding constrained soft Bellman policy.

---

**Inner process**. As $\bar{\mathcal{N}}^{[v]}(n) = \mathcal{V}$, learner $v$ knows the cost weight vector of all the learners. Given $\omega_c^{[v]}$, at inner iteration $k$, each learner $\bar{v}$ receives $\eta^{[\bar{v}']}(\omega_c^{[v]}, k)$ from neighbors and updates its estimate of the reward weight vector (embedded in $\eta^{[\bar{v}]}$). In specific, at line 4 in Algorithm 2, the first term (convex combination) on the right hand side encourages consensus and the second term (gradient) drives to the optimal solution. It is shown in Lemma 3 that each learner can reach $\eta^*(\omega_c^{[v]})$ at the rate of $O(\frac{1}{\sqrt{\log K}})$.

**Outer process**. Once obtaining $\bar{\eta}^{[v]}(\omega_c^{[v]})$, each learner $v$ faces three challenges in solving the outer problem: (i) the gradient of $F^{[v]}(\omega_c^{[v]}, \eta^*(\omega_c^{[v]}))$ cannot be obtained because the inner problem is not fully solved, i.e., learner $v$ does not find $\eta^*(\omega_c^{[v]})$ but its approximation $\bar{\eta}^{[v]}(\omega_c^{[v]})$, (ii) $\sum_{v' \neq v} F^{[v']}$ is unknown, and (iii) the local likelihood function $F^{[v]}$ is non-convex. To solve these challenges respectively, we propose three techniques: local gradient approximation (LGA), global gradient approximation (GGA), and local successive convex approximation (LSCA). By subtly designing these three approximations, our algorithm is guaranteed to converge to the set of stationary points. Notice that the second challenge also arises in the inner problem but we solve it in a simpler way because the inner problem is convex.

To approximate the local gradient $\nabla F^{[v]}(\omega_c, \eta^*(\omega_c))$, we propose the following LGA of learner $v$: $\bar{\nabla} F^{[v]}(\omega_c, \bar{\eta}(\omega_c)) = \sum_{\zeta^j \in \mathcal{D}^{[v]}} \sum_{t=0}^{\infty} \gamma^t \phi_c(s_t^j, a_t^j) - m^{[v]} E_{S,A}^{\pi_{\bar{\eta}(\omega_c);\omega_c}}[\sum_{t=0}^{\infty} \gamma^t \phi_c(S_t, A_t)]$, whose derivation is based on the strict convexity of $G(\eta; \omega_c)$ and $G^{[v]}(\eta; \omega_c)$. The derivation and the proof of strict convexity are both included in the Appendix. Moreover, it can be shown (in the Appendix) that the approximation error (defined in the Appendix) is upper bounded and reduces to zero as $K \to \infty$.

While learner $v$ can use LGA to get a sound approximation of $\nabla F^{[v]}$, in order to solve problem (3), it still needs the knowledge of the gradients $\sum_{v' \neq v} \nabla F^{[v']}$ of other learners to get the global gradient. Therefore, we leverage a GGA $\bar{\nabla}^{[v]}$ (lines 5-10) who aims to track the average global gradient $\frac{1}{N_L} \nabla F(\omega_c^{[v]}, \eta^*(\omega_c^{[v]}))$. However, $\nabla F(\omega_c^{[v]}, \eta^*(\omega_c^{[v]}))$ is not available due to the inaccessibility of $\eta^*(\omega_c^{[v]})$. Therefore, $\bar{\nabla}^{[v]}$ is designed to track the approximation $\frac{1}{N_L} \sum_{v'=1}^{N_L} \bar{\nabla} F^{[v']}(\omega_c^{[v']}, \bar{\eta}^{[v']}(\omega_c^{[v']}))$ of the average global gradient.

To tackle the non-convexity issue, learner $v$ solves a LSCA problem (line 11) and then follows two-step updates of the local cost weight vector $\omega_c^{[v]}$. LSCA proposes to solve a local convexification of problem (3). In specific, learner $v$ solves the problem $\arg\max_{\omega_c \in \Omega_c} \tilde{F}^{[v]}(\omega_c; \omega_c^{[v]})$ where $-\tilde{F}^{[v]}(\omega_c; \omega_c^{[v]})$ is a local strongly-convex surrogate of $-F(\omega_c, \eta^*(\omega_c))$ at $\omega_c^{[v]}$ satisfying the gradient consistency condition $\nabla \tilde{F}^{[v]}(\omega_c; \omega_c^{[v]}) = N_L \bar{\nabla}^{[v]}$. An attractive property of LSCA [27] is

that its fixed point exists and is a stationary point of functions with gradient $N_L \bar{\nabla}^{[v]}$ under certain conditions (in the Appendix).

Once obtaining $\tilde{\omega}_c^{[v]}$, learner $v$ executes two-step updates. The first update towards $\tilde{\omega}_c^{[v]}$ (line 12) drives learner $v$ to a stationary point of problem (3) and the second update (lines 13-14) encourages the consensus among different learners.

**Lemma 2.** *The result $\tilde{\omega}_c^{[v]}(n)$ at line 11 in Algorithm 1 is the optimal solution of a LSCA problem at $\omega_c^{[v]}(n)$ under the GGA $N_L \bar{\nabla}^{[v]}(n)$.*

**Lemma 3.** *(Convergence rate of the inner problem) Suppose Assumptions 1 and 2 hold and let $\alpha(k) = \frac{\bar{\alpha}}{k+1}$ where $\bar{\alpha}$ is a positive constant. Then, for every learner $v \in \mathcal{V}$ in Algorithm 2,*

$$||\bar{\eta}^{[v]}(\omega_c) - \eta^*(\omega_c)|| \le \frac{C_1^{[v]}}{\log K} + \frac{C_2^{[v]}}{\sqrt{\log K}},$$

*where $C_1^{[v]}$ and $C_2^{[v]}$ are positive constants whose expression can be found in the Appendix.*

We define the set of KKT points (also called stationary points) of problem (3)-(4) as [28] [29]: $\Omega_c^* \triangleq \{\omega_c^* \in \Omega_c : \max_{\omega_c \in \Omega_c}\{(\nabla F(\omega_c^*, \eta^*(\omega_c^*)))^\top (\omega_c - \omega_c^*)\} = 0\}$, and study the convergence property of the following metric [30]: $J(\omega_c(n)) \triangleq \max_{\omega_c \in \Omega_c}\{(\nabla F(\omega_c(n), \eta^*(\omega_c(n))))^\top (\omega_c - \omega_c(n))\}$.

**Theorem 1.** *(Asymptotic convergence of the outer problem) Suppose Assumptions 1, 2, and 3 hold. Let $\beta(n) \in (0,1)$, $\sum_{n=0}^{\infty} \beta(n) = +\infty$, and $\sum_{n=0}^{\infty} (\beta(n))^2 < +\infty$, then in Algorithm 1:*

$$(consensus): \quad \lim_{n \to \infty} \max_{v,v' \in \mathcal{V}} ||\omega_c^{[v]}(n) - \omega_c^{[v']}(n)|| = 0,$$

$$(convergence): \quad \limsup_{n \to \infty} J(\omega_c^{[v]}(n)) \le \frac{\bar{M}}{\sqrt{\log K}},$$

*where $\bar{M}$ is a positive constant whose existence is proved in the Appendix.*

Lemma 3 and Theorem 1 show that the distributed learners can achieve common values of $\omega_c$ and $\eta$. As the constrained soft Bellman policy is continuous in $(\omega_c, \eta)$ (proved in the Appendix), they will also achieve a common policy.

# 6 Simulation

This section presents two simulation examples. In the first example, the experts are programmed to follow the optimal policy under discrete state and action spaces. In the second example, the experts are humans, i.e., may not be optimal, and the states and actions are both continuous.

## 6.1 Synthetic grid world

We consider the grid world introduced in [11]. Different from paper [11]'s MDP, we implement a CMG on this environment where three experts perform motion planning from $s_0/s_0'/s_0''$ to $s_G/s_G'/s_G''$ while avoiding collision with obstacles and each other. The environment consists of a 9-by-9 grid of states, and the actions of each expert are to stay still, move up, down, left, right, or diagonally by one cell. The dynamics is stochastic as each action of an expert has 20% probability of failure, resulting in staying still. While the grid world looks small, the CMG has more than $387,000,000$ state-action pairs due to multiple experts. Each state-action pair produces a distance-relevant reward feature and the ground truth reward of each expert increases if the distance to its goal decreases. The true CMG (Figure 2a), from which the experts generate demonstration, includes constraints (i.e., obstacles) and the nominal MG does not. We use heat maps to show the relevant visitation frequency of each state and action pair. Each expert will terminate its motion if violating a constraint, i.e, colliding with an obstacle.

In our algorithm, there are four distributed learners where each learner can only respectively get 10, 20, 30, 40 demonstrated trajectories of the experts. Notice that the learners and experts are not the same entities, i.e., the learners stand aside observing the experts. The four learners jointly

recover the experts' policies by learning the constraints and the reward functions over an underlying communication network. The detailed simulation setup is included in the Appendix.

Furthermore, we use the following three baselines for comparisons:
**ME-greedy**: This method is introduced in [11] where a greedy method is used to estimate the constraints based on the ME framework.
**MCE-greedy**: We construct this method by extending ME-greedy to the same greedy method based on the MCE framework which is more suitable for stochastic environments.
**Centralized inverse constrained inverse reinforcement learning (C-ICRL)**: This method is a special case of D-ICRL where there is a centralized learner obtaining all the demonstration data.

Notice that ME-greedy and MCE-greedy assume the access to the ground truth rewards while C-ICRL does not have this assumption. All the three baselines are centralized learning where a single learner can get the total 100 demonstrated trajectories. Our algorithm is distributed learning and does not obtain the ground truth rewards, therefore, it solves a more difficult problem than the baselines do. However, the simulation shows that our algorithm can be on par with and even outperform the baselines in some aspects. In our bi-level algorithm, the outer loop is to estimate the constraints and the inner loop is to recover the corresponding policy by estimating the reward function. To reason about the performance of our algorithm, we use five metrics: cumulative rewards (CR), false positive rate (FPR), false negative rate (FNR), constraint violation rate (CVR), and reward function distance (RFD). The CR, commonly used in IRL and imitation learning literature [31, 32], illustrates the total reward collected by the three experts in an episode, FPR, introduced in [11], is the proportion of learned constraints that are not the ground truth constraints, FNR is the proportion of the ground truth constraints that are not learned, CVR [12] is the average percentage of the three experts' violating any constraint in an episode. Since the reward function is a linear combination of the reward features, we propose RFD to measure the distance between the learned reward weight vector $\omega_r$ and experts' reward weight vector $\omega_{r_E}$: $||\omega_r - \omega_{r_E}||/||\omega_{r_E}||$.

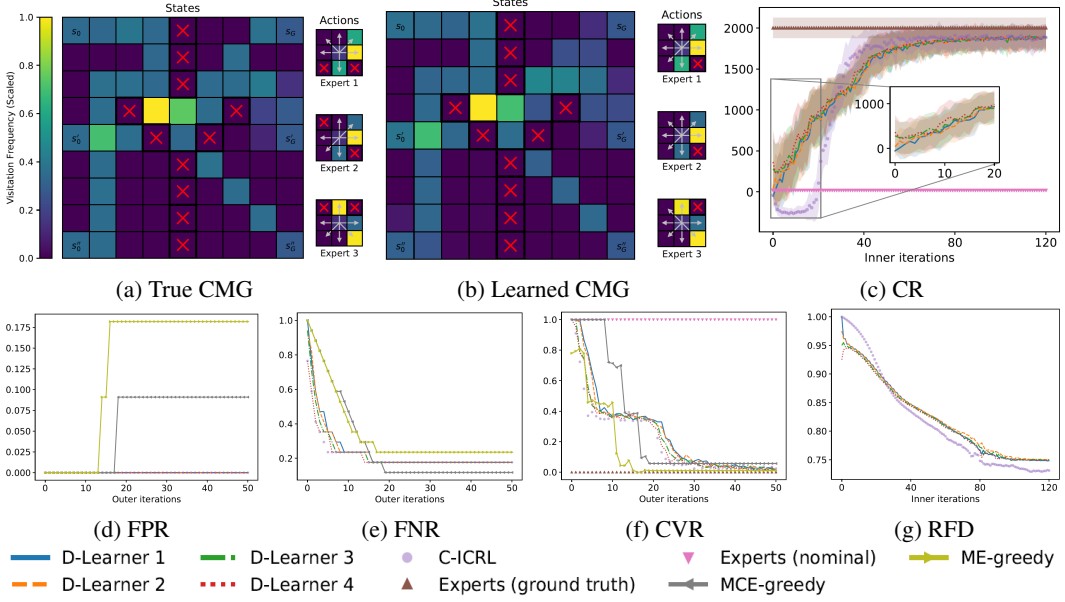

Figure 2: Algorithm performance on a synthetic grid world CMG. In subfigures 2a 2b, each state and action cell is colored according to the relevant visitation frequency and the red crosses represent the constraints. In subfigures 2c-2g, D-learners 1-4 are distributed learners in our algorithm. Because ME-greedy and MCE-greedy are not bi-level, we only include their curves against outer iterations.

Figure 2b shows the constraints learned by our algorithm (each learner recovers the same constraints). Several action constraints are not recovered due to the reason that the experts will barely take these actions even if they are feasible. Figures 2c 2f show that our learning algorithm can successfully imitate the experts' behavior according to CR and CVR. Figures 2d 2e show that D-ICRL

and C-ICRL have better FPR performance compared to the greedy-based methods while MCE-greedy has the best FNR performance. From Figure 2e and the small window in Figure 2c, we can see that the distributed learners can quickly achieve consensus even if they start from different initial conditions in both outer and inner loops. Figures 2c and 2g show the performance of inner process in the last outer iteration. Figure 2g shows that D-ICRL and C-ICRL can approach the ground truth reward functions and stabilize.

Table 1: Performance Comparisons. Here, D means distributed and NATR means no access to the (ground truth) rewards.

|  |  | D | NATR | CR | FPR | FNR | CVR | RFD |
|---|---|---|---|---|---|---|---|---|
| Experts (ground truth) |  | $N/A$ | $N/A$ | $2002.59 \pm 125.71$ | $N/A$ | $N/A$ | 0.000 | 0.000 |
| D-ICRL (our method) | D-Learner 1 | ✓ | ✓ | $1890.67 \pm 135.22$ | 0.00 | 0.18 | 0.013 | 0.748 |
|  | D-Learner 2 | ✓ | ✓ | $1873.07 \pm 162.14$ | 0.00 | 0.18 | 0.020 | 0.750 |
|  | D-Learner 3 | ✓ | ✓ | $1878.51 \pm 131.55$ | 0.00 | 0.18 | 0.020 | 0.749 |
|  | D-Learner 4 | ✓ | ✓ | $1884.85 \pm 167.16$ | 0.00 | 0.18 | 0.010 | 0.749 |
| C-ICRL |  | ✗ | ✓ | $1884.74 \pm 158.90$ | 0.00 | 0.18 | 0.013 | 0.731 |
| ME-greedy |  | ✗ | ✗ | $1162.24 \pm 132.66$ | 0.18 | 0.24 | 0.010 | $N/A$ |
| MCE-greedy |  | ✗ | ✗ | $1776.84 \pm 300.61$ | 0.09 | 0.12 | 0.057 | $N/A$ |
| Experts (nominal) |  | $N/A$ | $N/A$ | $18.64 \pm 3.02$ | $N/A$ | $N/A$ | 1.000 | 0.000 |

Table 1 shows the comparison result where each learner in our distributed algorithm can be on par with C-ICRL in all the five metrics. Our algorithm can outperform ME-greedy introduced in [11] in almost every metric except being on par with it in CVR. While MCE-greedy is centralized learning and assumes the access to the ground truth rewards, our method can still outperform it in three metrics, i.e., CR, FPR, and CVR.

## 6.2 Drones motion planning with obstacles

In the second example, we simulate in a physical setting with continuous state-action space. We build a simulator in Gazebo where each drone aims to reach the door in its diagonal direction while avoiding collisions. We first control the simulated drones to their target doors, record nine pairs of trajectories, and distribute four and five pairs to two learners respectively. The demonstration is shown in Figure 3b and each cell in the figure represents the constraint set indicated by a cost feature of each expert, i.e., $\mathcal{C}_j^{[i]}$. We define the reward feature of each state as its location multiplied with a coefficient. Figure 3c shows the learned constraints and trajectories.

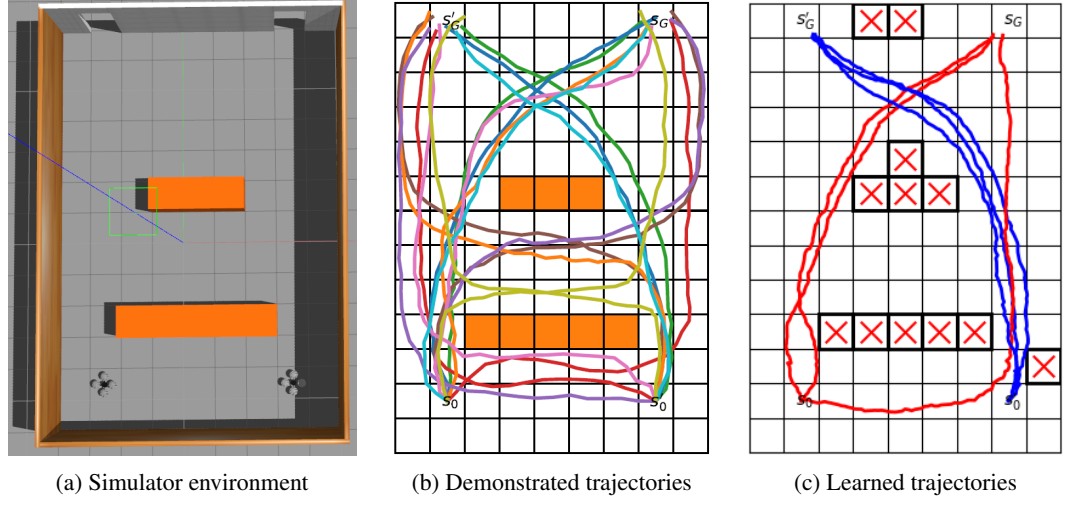

(a) Simulator environment     (b) Demonstrated trajectories     (c) Learned trajectories

Figure 3: Drones motion planning with obstacles

We keep the three metrics, FPR, FNR, and CVR in last example. Because the ground truth reward is not accessible in this physical setting, we cannot use RFD and we use success rate (SR) instead

of CR. The SR is defined as the percentage of the drones' successfully reaching the doors without any collision. We use the reward function learned from D-ICRL as the input reward function of ME-greedy. Because this environment is deterministic, MCE-greedy reduces to ME-greedy.

Table 2: Performance Comparisons.

| | | D | NATR | FPR | FNR | CVR | SR |
|---|---|---|---|---|---|---|---|
| D-ICRL | D-Learner 1 | ✓ | ✓ | 0.028 | 0.00 | 0.03 | 0.97 |
| | D-Learner 2 | ✓ | ✓ | 0.028 | 0.00 | 0.03 | 0.97 |
| C-ICRL | | × | ✓ | 0.018 | 0.00 | 0.02 | 0.98 |
| M(C)E-greedy | | × | × | 0.037 | 0.13 | 0.16 | 0.84 |

## 7  Discussion and future work

We propose D-ICRL, the first multi-expert-multi-learner MCE ICRL framework that is effective to CMG with both discrete and continuous environments. We employ MLE to estimate the constraints, use an MCE-based optimization problem to learn the corresponding reward function and policy, and derive our algorithm based on a novel distributed bi-level optimization theoretical framework. Experimental results show that D-ICRL can imitate the experts' behavior as well as recover the environmental constraints. Despite its benefits, the limitations are that (i) the learners assume that the reward/cost function is a linear combination of some features; (ii) the communication between learners is frequent. We will address the issues in the future.

## 8  Acknowledgements

This work is partially supported by National Science Foundation through grants CNS 1830390, ECCS 1846706, and ECCS 2140175. We would like to thank the reviewers for their insightful and constructive suggestions.

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
