# OpenReview forum: "Distributed Inverse Constrained Reinforcement Learning for Multi-agent Systems"
_NeurIPS.cc/2022/Conference — NeurIPS 2022 Accept_

### Official Review · Reviewer_7HCk · 2022-06-18

**Rating:** 6
**Confidence:** 4
**Soundness:** 4 excellent
**Presentation:** 2 fair
**Contribution:** 3 good

**Summary:**

This paper proposes an inverse RL problem that requires inferring both the rewards and constraints in a multi-agent system. Unlike previous multi-agent IRL work, this paper introduces a setting where sharing expert demonstrations between agents are not allowed. To solve the problem, this paper proposes a distributed bi-level learning algorithm that iteratively solves 1) an inner problem for learning rewards and imitation policy and 2) an outer problem for inferring constraints. This paper analyses the convergence rate of their algorithm in terms of finding the optimal solution to the inner problem and converging to the set of stationary points of the outer problem.

**Questions:**

1. In line 90, the paper mentions "we study the case where b = 0, i.e., hard constraint.", but later the paper estimates a budget b_wc in Equation (2). I believe the estimated b_wc is not equivalent to b, so the paper is solving a soft constraint problem. Please explain more about this budget.

2. In line 151, the paper defines the learner $\nu$’s local likelihood function as a discounted cumulative log-likelihood. Why do we need the discounting term here? This is the likelihood for a given trajectory j instead of state-action pairs.

3. This paper defines a bi-level optimization problem in formulas 3) and 4), but the following part has not introduced any insights or designed any algorithms for bi-level optimization.  It seems this paper just solves the inner and outer problems separately while the mainstream bi-level optimization algorithms often follow a single-level reformulation for better theoretical and practical properties. check [1]. Without these properties, what's the benefit of defining such a bi-level optimization problem?

4. The paper mentions that "the demonstration data are unable to be shared because of privacy concerns and communication burdens", but in algorithms 1 and 2, the agents are sharing information (the gradients and parameters) with each other. This information-sharing behavior happens in both the inner and the outer problems (is very frequent during learning) and thus it creates even larger communication burdens, which makes the motivation less significant. Have the authors thought about minimizing the sharing frequency?

5. I am wondering if it is necessary to learn a separate constraint function for each of the learner agents. What's the issue of learning a global constraint function? Since the information needs to be shared across the agents anyway.

6. I am wondering if the drone's motion planning environment is indeed a multi-agent environment. It seems the expert trajectories run across each other a lot. Will the drones crash into each other in this way?

7. In line 166. These citations have not mentioned constraints.

6. In algorithm 1, what does n+1/2 means?

[1] Risheng Liu, Jiaxin Gao, Jin Zhang, Deyu Meng, Zhouchen Lin: Investigating Bi-Level Optimization for Learning and Vision from a Unified Perspective: A Survey and Beyond. CoRR abs/2101.11517 (2021)

**Limitations:**

### limitations

Section 7 is for the conclusion and future work. The discussion about limitations is unsatisfying. I suggest including my concerns about the reward functions and learning settings (see the weaknesses about paper significance mentioned above) in the limitations. In fact, many works have been aware of these weaknesses and addressed them in many ways. I encourage the authors to explore more about it.

### negative societal impact
This paper has not discussed the negative societal impacts.


**Strengths And Weaknesses:**

### Strengths

**originality.** This paper extends the constraint IRL problem to the multi-agent domain, and more importantly, they assume the ground truth rewards are not available. To the best of my knowledge, previous works commonly assume one of these two dynamics is known (either assume the constraints are known or rewards are known) in order to infer the other one. This paper solves a very challenging problem, but it also gives me a feeling that maybe the problem itself is ill-posed since the IRL system can assign negative rewards in order to prevent some illegal movements. On the other hand, some constraint RL methods (e.g., Reward Constrained Policy Optimization [1]) augment the constraint signals to rewards by using the Lagrange multiplier. I mention this example since I believe the **learned** rewards can embed constraint signals unless you have other assumptions, for example, 1) the **system** rewards are fixed and we need constraints to explain the expert demonstrations (This is what Constraint IRL has). 2) the policy in maximum entropy IRL cannot assign zero probability to some actions no matter how bad the reward is, and thus we need some extra cost function in order to **ensure** hard constraints.

[1] Chen Tessler, Daniel J. Mankowitz, and Shie Mannor. Reward constrained policy optimization. In International Conference on Learning Representations ICLR. OpenReview.net, 2019.

**quality.** The quality of this paper is satisfying. The problem is generally well-defined. Under some reasonable assumptions (linear cost and reward functions), the bi-level objective ensures some important properties (e.g., local convexity in sub-problem). The proposed algorithm looks fine. This paper shows some theoretical results by analyzing the convergence rate of their methods.  The empirical simulation makes sense.


### Weaknesses

**clarity.** This paper defines a complicated problem, but it is not in a reader-friendly format. This paper splits the definition of experts and learners agents and put them into a multi-agent system. Following these definitions is difficult especially when some of the definitions are missing and some symbols are used before defining (see typos). I strongly recommend adding some comments to the algorithm 1 and 2, or at least tell the reader where to find their explanations (on page 6 I think). The overlapping subscripts in algorithm 1 (e.g. line 4 of the algorithm) is difficult to follow. I spend lots of effort understanding them, but I am still not sure whether I am right. See questions as follows.

typos:
1. lc anf lr in line 97 and line 99 are not defined.
2. I was confused by the \nu in line 93 and line 94, but then I find their meaning in the later sections when I read the part about communication networks.

**significance.** I am concerned about the significance of this work from the following perspectives.

- This algorithm assumes an open-box box environment with a known transition dynamic (Line 95, P and P_0 are the input of learners). The experiment uses two full-observable girdworld environments. More advanced benchmarks for constraint RL (e.g., safe-gym) have not been explored. Scaling the performance to solve the problem with high-dimensional and continuous state space will be difficult.

- This paper follows classic apprenticeship learning algorithms that restrict rewards to convex sets given by linear combinations of feature vectors. It is known that apprenticeship learning algorithms generally do not recover expert-like policies if the set is too restrictive, which is often the case for the linear subspaces used by feature expectation matching unless the feature functions are very carefully designed [2]. I think this is the reason why this paper adds an extra cost function to fill the gap.

[2] Jonathan Ho, and Stefano Ermon. Generative Adversarial Imitation Learning. In Neurips.

---

> ### Author Response · Authors · 2022-08-02
> **Response to Reviewer 7HCk**
>
> Thank you for your detailed and constructive feedback. We believe that this discussion can lead to a stronger paper. We address the comments below.
>
> $Weakness$ 1 mentioned in the originality: This paper solves a very challenging problem, but it also gives me a feeling that maybe the problem itself is ill-posed since the IRL system can assign negative rewards in order to prevent some illegal movements. On the other hand, some constraint RL methods (e.g., Reward Constrained Policy Optimization) augment the constraint signals to rewards by using the Lagrange multiplier. I mention this example since I believe the learned rewards can embed constraint signals unless you have other assumptions, for example, 1) the system rewards are fixed and we need constraints to explain the expert demonstrations (This is what Constraint IRL has). 2) the policy in maximum entropy IRL cannot assign zero probability to some actions no matter how bad the reward is, and thus we need some extra cost function in order to ensure hard constraints.
>
>
> $Answer$: It is insightful that you mention that (i) negative reward can be assigned to prevent illegal movements and (ii) the reward function can embed constraint signals. This is a very interesting idea and has achieved success in RL as you mention. However, this may be problematic in IRL.
>
> For (i), it is a little bit ambitious to just learn a reward function and hope it to successfully assign negative rewards to prevent illegal movements. We first try this method and the result is not good, especially for the obstacle avoidance. One reason is that the linear reward class is not powerful enough to explain such complicated behaviors and thus we need to additionally learn constraint. We agree that using more powerful models such as neural networks may alleviate this problem, however, as discussed in the next paragraph, even augmenting some constraint signals into the neural networks may still not produce satisfying results.
>
> For (ii), in the experiment section in [D1], the authors try this idea where they use a revised generative adversarial imitation learning (GAIL) algorithm with neural networks by augmenting the reward function with some constraint signals to compare with their methods and the performance of this GAIL revision is not very satisfying. Moreover, when we first try to design augmented reward features that contain the constraint information, we find it much more difficult than separately designing features for reward and constraint because we need to consider the effects of both the reward and constraint when designing such features.
>
> $Weakness$ 2: This paper defines a complicated problem, but it is not in a reader-friendly format. This paper splits the definition of experts and learners agents and put them into a multi-agent system. Following these definitions is difficult especially when some of the definitions are missing and some symbols are used before defining (see typos). I strongly recommend adding some comments to the algorithm 1 and 2, or at least tell the reader where to find their explanations (on page 6 I think). The overlapping subscripts in algorithm 1 (e.g. line 4 of the algorithm) is difficult to follow. I spend lots of effort understanding them, but I am still not sure whether I am right.
>
> $Answer$: Thank you for your suggestions. In order to improve paper readability, we include a graph to represent the relations of experts and learners in the appendix (Subsection 11.1) now. We follow your suggestion and solve the typo issues (highlighted in blue). For the description of Algorithms 1 and 2, we explain these two algorithms on page 6 step by step and all the notations in the algorithms are explained on page 6 and Sections 2-4. For example, the notations in line 4 in Algorithm 1 are explained in line 7 in Algorithm 2 and Lemma 1.

---

> > ### Author Response · Authors · 2022-08-02
> > **Response to Reviewer 7HCk (continued)**
> >
> > $Weakness$ 3: This algorithm assumes an open-box environment with a known transition dynamic (Line 95, P and $P_0$ are the input of learners). The experiment uses two full-observable girdworld environments. More advanced benchmarks for constraint RL (e.g., safe-gym) have not been explored. Scaling the performance to solve the problem with high-dimensional and continuous state space will be difficult.
> >
> > $Answer$: Thank you for mentioning this. We agree that knowing $P_0$ and $P$ will lead to a simpler RL algorithm. However, in IRL, our focus is on how to imitate the experts and IRL literature usually assume that the learner can get access to an RL oracle [D2] [D3]. The assumption of knowing the dynamics can be easily relaxed by using some more advanced RL algorithms and use trajectory samples to get an unbiased estimate of reward feature expectation and cost expectation. Therefore, this assumption does not affect the algorithm design and analysis of our IRL problem as IRL assumes the access to an RL oracle.
> >
> > In our second experiment, the state-action space is continuous (lines 294-295) and the grid only represents the possible discrete constraints. Therefore, our algorithm can be used in continuous state-action spaces. In fact, in the second experiment, the learners do not know the dynamics nor the whole state-action space (i.e., black box) as we use a revised soft Q-learning algorithm.
> >
> > $Weakness$ 4: This paper follows classic apprenticeship learning algorithms that restrict rewards to convex sets given by linear combinations of feature vectors. It is known that apprenticeship learning algorithms generally do not recover expert-like policies if the set is too restrictive, which is often the case for the linear subspaces used by feature expectation matching unless the feature functions are very carefully designed [2]. I think this is the reason why this paper adds an extra cost function to fill the gap.
> >
> > $Answer$: Your comment is insightful and precise. By using the extra constraint learning, the experiment shows that our algorithm can successfully imitate the experts' behaviors. Even with the assumption of linear combinations, it requires more than 12 pages to prove the main theorem. We believe that this setting is challenging enough for a publication. In the future work, we will investigate approaches that relax this assumption as discussed in Section 7.
> >
> > $Q1$: In line 90, the paper mentions "we study the case where b = 0, i.e., hard constraint.", but later the paper estimates a budget $b_{w_c}$ in Equation (2). I believe the estimated $b_{w_c}$ is not equivalent to b, so the paper is solving a soft constraint problem. Please explain more about this budget.
> >
> > $Answer$: Thank you for mentioning this constraint. The experts are solving a hard constraint problem with respect to the environment (b=0). The learners are solving a hard constraint problem with respect to the experts (the constraint in (2)). The intuition behind this is that the learners do not know the ground truth constraints so that they cannot directly solve the hard constraint problem with respect to the environment. However, the learners can indirectly solve the hard constraint problem with respect to the environment as long as they match the experts. In (2), we cannot directly use $b=0$ because this may cause infeasibility as discussed in Remark 2. Even though $b_{\omega_c}$ is not equivalent to $b$, when the learners can match the experts with respect to $b_{\omega_c}$, they can satisfy $b=0$ under the ground truth cost in the environment because they match the experts' behavior and the experts satisfy $b=0$. Of course, we need to choose appropriate cost feature to guarantee this. For example, in the second experiment, the union of the constraint sets indicated by all the cost features occupy the whole state space (lines 298-299) so that any state constraint in the system can be captured by the cost feature.

---

> > > ### Author Response · Authors · 2022-08-02
> > > **Response to Reviewer 7HCk (continued)**
> > >
> > > $Q2$: In line 151, the paper defines the learner’s local likelihood function as a discounted cumulative log-likelihood. Why do we need the discounting term here? This is the likelihood for a given trajectory j instead of state-action pairs.
> > >
> > > $Answer$: Thank you for mentioning the definition of the likelihood function. This definition follows the idea of the literature [D4] [D5]. For a finite trajectory $\zeta=s_0,a_0,\cdots,s_T,a_T$, the log likelihood is $\log P(\zeta)=\sum_{t=0}^{T}[\log P(s_t|s_{t-1},a_{t-1})\pi(a_t|s_t)]=\sum_{t=0}^{T}[\log P(s_t|s_{t-1},a_{t-1})+\log\pi(a_t|s_t)]$ where $P(s_0|s_{-1},a_{-1})$ is $P_0$. The thesis [D4] formulates the log likelihood of the trajectory $\zeta$ as $\sum_{t=0}^{T}\log\pi(a_t|s_t)$ where the dynamics terms are removed since the dynamics is given and set. In our problem, the trajectory is infinite-horizon. We cannot define the log likelihood as $\sum_{t=0}^{\infty}\log\pi(a_t|s_t)$ because this is the sum of infinite terms which may not be finite. Therefore, we follow the trick of defining the infinite-horizon causal entropy in [D5] and add the discount factor to ensure that the log likelihood is well defined. This is similar to the definition of cumulative reward in line 86. Notice that the discount factor can be chosen arbitrarily close to $1$.
> > >
> > > $Q3$: This paper defines a bi-level optimization problem in formulas 3) and 4), but the following part has not introduced any insights or designed any algorithms for bi-level optimization. It seems this paper just solves the inner and outer problems separately while the mainstream bi-level optimization algorithms often follow a single-level reformulation for better theoretical and practical properties. check [1]. Without these properties, what's the benefit of defining such a bi-level optimization problem?
> > >
> > > $Answer$: Thank you for mentioning the algorithm design. Our algorithm solves the bi-level problem through a double-loop way similar to [D6] [D7]. In fact, our algorithm does not solve the outer and inner problems separately because the result of the inner problem is needed to solve the outer problem and the inner problem is also parameterized by the decision variable of the outer problem. Explained in lines 178-181, the algorithm first solves the inner problem, and then uses the result of the inner problem to solve the outer problem. Once the parameter of the outer problem is updated, we have a new inner problem because the inner problem is parameterized by the parameter of the outer problem. Detailed explanations of the algorithm design can be found on pages 5-6.
> > >
> > > We appreciate that you mention the single-level reformulation literature [D8]. In [D8], the authors introduce a "best-response-based single-level reformulation" which tries to first find the best response $\eta^{\ast}(\omega_c)$ of the inner problem given the decision variable $\omega_c$ of the outer problem. Actually, our algorithm adopts the same idea. The authors call this "single-level" in the sense that the bi-level problem becomes single-level once we know the optimal solution of the inner problem. Therefore, they call it ``best-response-based" single-level reformulation. Moreover, in Algorithms 2,3,4 in [D8], the inner problem is required to be solved for either $T$ iterations (Algorithms 2,3) or to an $\epsilon$-optimal solution (Algorithm 4) before solving the outer problem. This shares the idea of our algorithm because our algorithm also first solves the inner problem for $K-1$ rounds to get an approximation of the optimal solution of the inner problem and then uses this approximation to solve the outer problem.

---

> > > > ### Author Response · Authors · 2022-08-02
> > > > **Response to Reviewer 7HCk (continued)**
> > > >
> > > > $Q4$: The paper mentions that "the demonstration data are unable to be shared because of privacy concerns and communication burdens", but in algorithms 1 and 2, the agents are sharing information (the gradients and parameters) with each other. This information-sharing behavior happens in both the inner and the outer problems (is very frequent during learning) and thus it creates even larger communication burdens, which makes the motivation less significant. Have the authors thought about minimizing the sharing frequency?
> > > >
> > > > $Answer$: Thank you for mentioning the communication issue. We agree that the sharing of gradients and parameters is frequent, which is standard in distributed machine learning literature [D9] [D10]. Due to the privacy concern, the local learner cannot share its private data (e.g., local demonstration and local estimates of reward feature expectation and cost expectation) but can share the corresponding parameters to help protect the privacy. This is also how federated learning solves the privacy issue [D11]. In federated learning, the data of each node is needed to help learn a good global model while the local data of each node cannot be directly shared to the server due to privacy. Therefore, each node will transmit its gradients and parameters to the server to help the learning instead of its local data. Moreover, even if the communication is frequent, the overall communication burden can be reduced given that the gradients and parameters have much smaller dimensions compared to the demonstration data. Consider the flight queue control problem in [D5] where the demonstration set has 2,000 trajectories and the length of each trajectory is 50. Each state-action pair has 5 components, thus the total communication cost for the demonstration data is 500,000. The feature they choose has 5 components. As they do not consider constraints, for a fair comparison, we suppose the feature has 10 components. The total iteration number is less than 1,000, thus the total communication cost for the parameters is less than 10,000, which is only 2\% of the communication cost of the demonstration data.
> > > >
> > > > It is a very interesting direction to investigate reducing the communication frequency. An idea is event-triggering communication where each learner only communicates when the difference between its local parameters and the neighbors' parameters received last time is above a threshold [D12].
> > > >
> > > > $Q5$: I am wondering if it is necessary to learn a separate constraint function for each of the learner agents. What's the issue of learning a global constraint function? Since the information needs to be shared across the agents anyway.
> > > >
> > > > $Answer$: Thank you for mentioning the constraint learning. Each learner has a local estimate of the global cost function. As each learner only knows partial information, its initial local estimate of the global cost function is different from other learners'. However, with the help of the convex combination (line 14 in Algorithm1), the learners' local estimates of the global cost function will achieve consensus (proved in Theorem 1).
> > > >
> > > > $Q6$: I am wondering if the drone's motion planning environment is indeed a multi-agent environment. It seems the expert trajectories run across each other a lot. Will the drones crash into each other in this way?
> > > >
> > > > $Answer$: Thank you for mentioning this. The drone experiment is a multi-agent system and they consider avoiding collisions. Their trajectories run across each other but they reach the locations at different times, thus there is no collision.
> > > >
> > > > $Q7$: In line 166. These citations have not mentioned constraints.
> > > >
> > > > $Answer$: Yes, you are right. These reference does not consider constraints but we can follow the idea to solve our constrained problem. We include the detailed proof of Lemma 1 in the appendix (Subsection 9.2).
> > > >
> > > > $Q8$: In algorithm 1, what does n+1/2 means?
> > > >
> > > > $Answer$: The parameter $\omega_c^{[v]}(n+1/2)$ is an intermediate parameter used to update to $\omega_c^{[v]}(n+1)$ from $\omega_c^{[v]}(n)$. We use $n+1/2$ because it is between $n$ and $n+1$.
> > > >
> > > > $Limitations$: Section 7 is for the conclusion and future work. The discussion about limitations is unsatisfying. I suggest including my concerns about the reward functions and learning settings (see the weaknesses about paper significance mentioned above) in the limitations. In fact, many works have been aware of these weaknesses and addressed them in many ways. I encourage the authors to explore more about it.
> > > >
> > > > $Answer$: We agree that your concerns are important and insightful and we follow the suggestion and include the concerns in section 7 (highlighted in blue). We will also investigate approaches that relax the linear combination assumption and reduce the communication frequency.

---

> > > > > ### Author Response · Authors · 2022-08-02
> > > > > **Response to Reviewer 7HCk (continued)**
> > > > >
> > > > > [D1] Anwar, Usman and Malik, Shehryar and Aghasi, Alireza and Ahmed, Ali, “Inverse constrained reinforcement learning,” International Conference on Machine Learning, pp. 7390–7399, 2021.
> > > > >
> > > > > [D2] Abbeel, Pieter, and Andrew Y. Ng, ``Apprenticeship learning via inverse reinforcement learning," International conference on Machine Learning, pp. 1--8. 2004.
> > > > >
> > > > > [D3] Arora, Saurabh, and Prashant Doshi, ``A survey of inverse reinforcement learning: Challenges, methods and progress," Artificial Intelligence, vol. 297, p. 103500, 2021.
> > > > >
> > > > > [D4] Ziebart, Brian D, ``Modeling purposeful adaptive behavior with the principle of maximum causal entropy," Carnegie Mellon University, 2010.
> > > > >
> > > > > [D5] Zhou, Zhengyuan and Bloem, Michael and Bambos, Nicholas, ``Infinite time horizon maximum causal entropy inverse reinforcement learning," IEEE Transactions on Automatic Control, vol. 63, no. 9, pp. 2787-2802, 2017.
> > > > >
> > > > > [D6] Ghadimi, Saeed, and Mengdi Wang, ``Approximation methods for bilevel programming," arXiv preprint arXiv:1802.02246, 2018.
> > > > >
> > > > > [D7] Ji, Kaiyi, Junjie Yang, and Yingbin Liang, ``Bilevel optimization: Convergence analysis and enhanced design," International Conference on Machine Learning, pp. 4882-4892, 2021.
> > > > >
> > > > > [D8] Liu, Risheng and Gao, Jiaxin and Zhang, Jin and Meng, Deyu and Lin, Zhouchen, ``Investigating bi-level optimization for learning and vision from a unified perspective: A survey and beyond," IEEE Transactions on Pattern Analysis and Machine Intelligence, no. 1, pp. 1-1, 2021.
> > > > >
> > > > > [D9] Zhang, Kaiqing, Zhuoran Yang, Han Liu, Tong Zhang, and Tamer Basar, ``Fully decentralized multi-agent reinforcement learning with networked agents," International Conference on Machine Learning, pp. 5872-5881, 2018.
> > > > >
> > > > > [D10] Luo, Ping and Xiong, Hui and Lu, Kevin and Shi, Zhongzhi, ``Distributed classification in peer-to-peer networks," ACM SIGKDD International Conference on Knowledge Discovery and Data Mining, pp. 968-976, 2007.
> > > > >
> > > > > [D11] Yin, Xuefei and Zhu, Yanming and Hu, Jiankun, ``A comprehensive survey of privacy-preserving federated learning: A taxonomy, review, and future directions," ACM Computing Surveys, vol. 54, no. 6, pp. 1-36, 2021.
> > > > >
> > > > > [D12] Nowzari, Cameron and Garcia, Eloy and Cortes, Jorge, ``Event-triggered communication and control of networked systems for multi-agent consensus," Automatica, vol. 105, pp. 1-27, 2019.

---

> > > > > > ### Comment · Reviewer_7HCk · 2022-08-10
> > > > > > **Thanks for your response.**
> > > > > >
> > > > > > I have gone through all the responses. The authors have done a nice job! Some of the concerns are resolved while the authors also admit some limitations, but at least everything is clear. The paper is generally in a good shape. I do encourage the authors to simplify some notations and add more comments to the main algorithm. Such efforts are not just for publication but also for a larger impact on the field.
> > > > > >
> > > > > > I can increase my scores.

---

### Official Review · Reviewer_AoHD · 2022-07-11

**Rating:** 6
**Confidence:** 3
**Soundness:** 3 good
**Presentation:** 2 fair
**Contribution:** 3 good

**Summary:**

This study proposes an effective learning approach for “distributed inverse constrained reinforcement learning” (D-ICRL). The current study solves this targeted problem by using a bi-level distributed optimization approach in estimating the constraints of the Markov game, reward, and the optimum policy of the cooperating expert agents.

**Questions:**

1. Each distributed learner tries to learn the policies for all agents and estimate the cost function while utilizing the given demonstration data? It would be nice to graphically represent the target parameters that each learner tries to estimate and the scope of demonstrating data that is available for each learning agent.

2. Please explain in detail how the problem (1) is converted into (2) and (3). The proposed bi-level formulation is proposed by the authors and thus is considered to be novel or this process is a natural and straightforward derivation induced while transforming the centralized constrained optimization into the distributed dual formulation. If the bi-level formulation has unique technical novelty, the author should explain the derivation procedure and intuition behind it, otherwise, the current paper should not highlight the bi-level formulation part.



====Post Rebuttal========
Thanks for the detailed reply. I adjusted the score because the answer answered all of my questions. The reason that I cannot provide a more positive evaluation is that I don't fully understand the mathematical details. But, I believe this paper can contribute to the community.

**Limitations:**

The paper presentation is not clear. Instead of directly formulating the distributed version of constrained inverse multi-agent reinforcement learning, it would be nice to first precisely formulate the centralized version of ICRL.

The experiment section is too short and just provides numerical results without explaining the meaning and significance of the results.

The theoretical analysis looks rigorous, however, the meaning of such long results is not well explained.


**Strengths And Weaknesses:**

<Strength>
The proposed paper aims to tackle a challenging problem that has never been dealt with before.


<Weakness>

•	Motivation for having multiple learners is not clear.

•	Problem formulation for Distributed Inverse Constrained Reinforcement Learning for Multi-agent Systems is not very clear. As a result, the derivation procedure of distributed algorithm is also unclear.

•	Experiment setup is too simple and hard to evaluate in supporting the effectiveness of the proposed model

---

> ### Author Response · Authors · 2022-08-02
> **Response to Reviewer AoHD**
>
> Thank you for your detailed and insightful review. Discussing these points will help us improve and clarify our work. We address the comments below.
>
> $Weakness$ 1: Motivation for having multiple learners is not clear.
>
> $Answer$: Thank you for mentioning the motivation. The motivation of multiple learners is mentioned in lines 23-25 and we would like to further explain it. There is a large literature on distributed machine learning in multi-agent systems [C1] where training data is distributed to a group of learners and cannot be shared, and the learners communicate with each other to collectively solve learning tasks of interest. Distributed IRL falls under the umbrella of distributed machine learning. For example, consider the patroller-evader problem introduced in [C2] where there are two patrollers patrolling around an area and there are multiple evaders hidden outside the patrolled area which want to learn the patrolling policy and evade the area without being detected. The patrollers are the experts and the evaders are the learners. Each evader has its own observations of the patrollers and do not want to share the raw data. To better learn the patrolling policy, they communicate with each other and perform collaborative learning.
>
> $Weakness$ 2: Problem formulation for Distributed Inverse Constrained Reinforcement Learning for Multi-agent Systems is not very clear. As a result, the derivation procedure of distributed algorithm is also unclear.
>
> $Answer$: Thank you for mentioning this. In the problem formulation (Section 4), we derive our problem formulation based on the well-established maximum causal entropy inverse reinforcement learning literature [C3] [C4]. The core part of our derivation is Lemma 1 which explains that how problem (1) combined with problem (2) is transformed into problem (3)-(4). We omit the details of the derivation because the details are fully explained in the literature [C3] [C4]. In the algorithm part (Section 5), we explain how we design the algorithm to solve the problem (3)-(4) in lines 178-181 and page 6. This algorithm design of first solving the inner problem and thus using the results to solve the outer problem follows the idea of [C5] [C6] and is an intuitive and natural way to solve bi-level optimization problems. We would like to elaborate the algorithm design here and elaborate the derivation of the problem formulation in the response to the Question 2.
>
> The algorithm design follows the idea of [C5] [C6] and uses a double-loop style to solve the bi-level problem (3)-(4). The inner problem (4) is parameterized by the decision variable $\omega_c$ of the outer problem and the outer problem needs the optimal solution $\eta^{\ast}(\omega_c)$ of the inner problem to be fully defined. Therefore, a natural and intuitive way [C5] [C6] to solve the bi-level problem (3)-(4) is that:  firstly, given an $\omega_c$, we solve the inner problem and get an approximation $\bar{\eta}(\omega_c)$ of the optimal solution of the inner problem (lines 3-4 in Algorithm 1); secondly, we use this approximation to find an approximation gradient of the outer problem and then update the outer problem to get a new $\omega_c$ (lines 5-14 in Algorithm 1). We iterate these two processes until convergence and we can prove the convergence to the stationary point (Theorem 1). Notice that in the algorithm, there are two loops: inner loop and outer loop.
>
> $Weakness$ 3: Experiment setup is too simple and hard to evaluate in supporting the effectiveness of the proposed model.
>
> $Answer$: Thank you for mentioning the experiment. The experiment setup satisfies all the requirements and assumptions imposed in Section 2. The experiment results show that the distributed learners converge to consensus in both reward function and constraint which validates the theory developed in section 5. What's more, the experiment results show that the learners (i) successfully identify the constraints and avoid them; (ii) successfully imitate the experts' behavior according to metrics like cumulative reward and success rate. Compared to the baselines (lines 255-261), our algorithm can achieve the same and even better performance even if it is distributed and dose not require ground truth reward. Moreover, the grid world experiment is widely used in IRL literature [C7] [C8] [C9] and our first grid world setup is borrowed from paper [C7]. The second experiment has continuous state-action space (lines 236) and it simulates a practical real-world problem.

---

> > ### Author Response · Authors · 2022-08-02
> > **Response to Reviewer AoHD (continued)**
> >
> > $Q1$: Each distributed learner tries to learn the policies for all agents and estimate the cost function while utilizing the given demonstration data? It would be nice to graphically represent the target parameters that each learner tries to estimate and the scope of demonstrating data that is available for each learning agent.
> >
> > $Answer$: Yes, your understanding is exactly correct and it is a very constructive suggestion to use a graph to better illustrate the relations between experts and learners. Due to the page limit, we could not include the graph in the paper but we include the graph in the appendix (Subsection 11.1) now. Basically, each learner only has a portion of all the demonstrations of the experts and wants to use communication to collaboratively learn the reward weight vector $\omega_r$ and cost weight vector $\omega_c$. Once they know these two parameters, they can find the constrained soft Bellman policy using existing algorithms such as soft Q-learning and soft actor critic.
> >
> > $Q2$: Please explain in detail how the problem (1) is converted into (2) and (3). The proposed bi-level formulation is proposed by the authors and thus is considered to be novel or this process is a natural and straightforward derivation induced while transforming the centralized constrained optimization into the distributed dual formulation. If the bi-level formulation has unique technical novelty, the author should explain the derivation procedure and intuition behind it, otherwise, the current paper should not highlight the bi-level formulation part.
> >
> > $Answer$: First, we would like to point out that even the centralized version of the bi-level problem is novel as no one formulates this before. The derivation (lines 158-167) of this bi-level problem is based on reference [C3] [C4] and consists of two parts: notions and notations introduction (lines 158-162) and Lemma 1 (lines 163-165). Lemma 1 provides the gradient needed in Algorithm 1 and we do not take much credit for Lemma 1 as mentioned in lines 166-167. The intuition behind this bi-level formulation is inspired by hyperparameter learning [C6] and an IRL paper [C10]. Paper [C10], inspired by the hyperparameter learning, interprets IRL as a bi-level problem where the outer level is to learn the reward function and the inner level is to learn the corresponding policy. In our case, we treat the constraint as a hyperparameter in our learned environment and we need to recover the corresponding reward function and policy given current constraint. Therefore, in our problem, the outer level is to learn the constraint and the inner level is to learn the corresponding reward function and policy. We really appreciate that you point this out and we include this intuition in the appendix (Subsection 11.2) now due to the page limit of the paper.
> >
> > Below, we explain the problem formulation in detail. The formulation follows three steps:
> >
> > Step 1: We use a maximum likelihood estimation (MLE) problem to learn the constraint  where the policy is parameterized by the cost weight vector $\omega_c$ (problem (1)). To solve this MLE problem, we need to find a parametric policy model for $\pi_{\omega_c}$. In specific, we only know that the policy is parameterized by $\omega_c$, but we do not know how it is parameterized.
> >
> > Step 2: To find a parametric model, we use an optimization problem based on maximum causal entropy (MCE) scheme (problem (2)). Notice that problem (2) is parameterized by $\omega_c$, thus the optimal solution is also parameterized by $\omega_c$. The optimal solution of problem (2) is the parametric policy model we want to find. However, problem (2) cannot be directly solved, so that we use its dual problem. In Lemma 1, we show that the optimal policy of problem (2) is the constrained soft Bellman policy parameterized by $\eta^{\ast}(\omega_c)$ where $\eta^{\ast}(\omega_c)$ is the optimal solution of the dual problem of problem (2). We omit a lot of details here because this strictly follows the well-established MCE literature [C3] [C4].
> >
> > Step 3: From the last two steps, we know that we need to first solve problem (2) and use its optimal solution as the parametric policy model to solve problem (1). From Lemma 1, we know that we can find the optimal solution of problem (2) once we find the optimal solution $\eta^{\ast}(\omega_c)$ of the dual problem of (2). Thus, we can derive problem (3)-(4) where the inner problem (4) is to find $\eta^{\ast}(\omega_c)$ and once we know $\eta^{\ast}(\omega_c)$, we can find the policy model used in the outer problem (3) and solve the outer problem.
> >
> > The algorithm design follows this idea. We need to first solve the inner problem to get $\eta^{\ast}(\omega_c)$ through \emph{Inner process} and use the obtained result to solve the outer problem through outer process.

---

> > > ### Author Response · Authors · 2022-08-02
> > > **Response to Reviewer AoHD (continued)**
> > >
> > > $Limitation$ 1: The paper presentation is not clear. Instead of directly formulating the distributed version of constrained inverse multi-agent reinforcement learning, it would be nice to first precisely formulate the centralized version of ICRL.
> > >
> > > $Answer$: We directly formulate the distributed problem because of the page limit. We really appreciate that you point it out and we include the derivation of the centralized problem and how it is decomposed into the distributed problem here and in the appendix (Subsection 11.3).
> > >
> > > The global demonstration set D has m trajectories $\zeta^j$ where $j=1,\cdots,m$. The global log likelihood of D is $F(\omega_c)=\sum_{j=1}^m\sum_{t=0}^{\infty}\gamma^t \ln \pi_{\omega_c}(a_t^j|s_t^j)$. To solve this MLE problem, we need to find a parametric policy model of $\pi_{\omega_c}$. To find such a model, we use an optimization problem (2) based on MCE scheme. Notice that this optimization problem (2) is parameterized by $\omega_c$, thus its optimal solution is also parameterized by $\omega_c$. Its optimal solution is the policy model we want in the likelihood function $F(\omega_c)$. Following well-established literature [C3] [C4], we can see that the optimal solution of this optimization problem is the constrained soft Bellman policy $\pi_{\eta^{\ast}(\omega_c);\omega_c}$, where $\eta^{\ast}(\omega_c)$ is the optimization solution of the problem $\min_{\eta} G(\eta;\omega_c)$. To fully define the MLE problem, we also need the optimal solution $\eta^{\ast}(\omega_c)$ of the problem $\min_{\eta} G(\eta;\omega_c)$. Therefore, we formulate the MLE problem as a bi-level optimization problem: $\max_{\omega_c} F(\omega_c,\eta^{\ast}(\omega_c))=\sum_{j=1}^m\sum_{t=0}^{\infty}\gamma^t \ln \pi_{\eta^{\ast}(\omega_c);\omega_c}(a_t^j|s_t^j)$ s.t. $\eta^{\ast}(\omega_c)=\mathop{\arg\min}_{\eta} G(\eta;\omega_c)$. Now, we finish the derivation of the centralized bi-level problem.
> > >
> > > As no learner knows the global demonstration data $\mathcal{D}$ and no learner can formulate $F$ and $G$, the centralized bi-level problem cannot be directly solved. Therefore, we need to decompose the centralized problem into an equivalent distributed problem that the learners can solve even if each learner only knows its local demonstration. Therefore, we decompose $F$ into multiple $F^{[v]}$ and $G$ into multiple $G^{[v]}$. In Remark 3, we show that the distributed problem (3)-(4) is equivalent to the centralized bi-level problem we derive here.
> > >
> > > $Limitation$ 2: The experiment section is too short and just provides numerical results without explaining the meaning and significance of the results.
> > >
> > > $Answer$: Thank you for mentioning this. In the first experiment, we provide explanation of the results in Figure 1 (lines 278-287) and Table 1 (lines 288-292). Basically, the explanation of Figure 1 shows that the distributed learners achieve consensus in both reward functions and constraints which validates Lemma 3 and Theorem 1. It also discusses that our distributed learners can successfully imitate the experts according to the defined metrics. The explanation also analyzes the reason of why some constraints are not recovered. What's more, it compares the performance difference between the gradient-based methods (D-ICRL and C-ICRL) and greedy-based methods (ME-greedy and MCE-greedy). The explanation of Table 1 shows the significance in the sense that our algorithm can be on par with and even outperform the baselines even if it is distributed and does not require ground truth reward. We do not explain the results of the second experiment due to page limit and the reason that the insights behind the results of this experiment are similar to the first one. Moreover, we would like to mention that this paper focuses more on the theory part and thus the derivation of the problem formulation, algorithm design, and theoretical analysis occupy the most pages.

---

> > > > ### Author Response · Authors · 2022-08-02
> > > > **Response to Reviewer AoHD (continued)**
> > > >
> > > > $Limitation$ 3: The theoretical analysis looks rigorous, however, the meaning of such long results is not well explained.
> > > >
> > > > $Answer$: We include some brief explanation of the results in lines 230-232 due to page limit and we now include more detailed explanations here and in the appendix (Subsection 11.4).
> > > >
> > > > In our setting, each distributed learner only has a portion of the global demonstration data set and one goal of our distributed algorithm is that the distributed learners can learn as good as a centralized learner who obtains the global data set even if the distributed learners do not share their local demonstration set. In Lemma 3, it is shown that given a cost feature estimate $\omega_c$, the distributed learners can converge to a consensus and the consensus is the optimal solution of the inner problem, i.e., their consensus is the best solution the centralized learner can achieve. In Theorem 1, it is shown that the distributed learners can achieve consensus on the cost weight $\omega_c$ which belongs to the stationary point set of the outer problem. Combined Lemma 3 and Theorem 1, we can see that the distributed learners will converge to the consensus on both the reward function and cost function where the learned cost weight belongs to the stationary point set of the outer problem and the learned reward weight is the optimal solution of the inner problem. Given that the outer problem is non-convex and the inner problem is convex, this convergence to the stationary point set of the outer problem and the optimal solution of the inner problem is strong.
> > > >
> > > > [C1] Stone, Peter and Veloso, Manuela, ``Multiagent systems: A survey from a machine learning perspective," Autonomous Robots, vol. 8, no. 3, pp. 345-383, 2000.
> > > >
> > > > [C2] Bogert, Kenneth and Doshi, Prashant, ``Multi-robot inverse reinforcement learning uner occlusion with interactions," International Conference on Autonomous Agents and Multi-agent Systems, pp. 173-180, 2014.
> > > >
> > > > [C3] Ziebart, Brian D and Bagnell, J Andrew and Dey, Anind K, ``Modeling interaction via the principle of maximum causal entropy." International Conference on Machine Learning, pp. 1255-1262, 2010.
> > > >
> > > > [C4] Zhou, Zhengyuan and Bloem, Michael and Bambos, Nicholas, ``Infinite time horizon maximum causal entropy inverse reinforcement learning." IEEE Transactions on Automatic Control, vol. 63, no. 9, pp. 2787-2802, 2017.
> > > >
> > > > [C5] Ghadimi, Saeed, and Mengdi Wang. ``Approximation methods for bilevel programming." arXiv preprint arXiv:1802.02246, 2018.
> > > >
> > > > [C6] Ji, Kaiyi, Junjie Yang, and Yingbin Liang. ``Bilevel optimization: Convergence analysis and enhanced design." International Conference on Machine Learning , pp. 4882-4892. 2021.
> > > >
> > > > [C7] Scobee, Decter RR, and S. Shankar Sastry. ``Maximum likelihood constraint inference for inverse reinforcement learning." International Conference on Learning Representations, 2019.
> > > >
> > > > [C8] Reddy, Tummalapalli Sudhamsh and Gopikrishna, Vamsikrishna and Zaruba, Gergely and Huber, Manfred, `` Inverse reinforcement learning for decentralized non-cooperative multiagent systems," IEEE International Conference on Systems, Man, and Cybernetics, pp. 1930-1935, 2012.
> > > >
> > > > [C9] Wang, Xingyu and Klabjan, Diego, ``Competitive multi-agent inverse reinforcement learning with sub-optimal demonstrations," International Conference on Machine Learning, pp. 5143-5151, 2018.
> > > >
> > > > [C10] Das, Neha and Bechtle, Sarah and Davchev, Todor and Jayaraman, Dinesh and Rai, Akshara and Meier, Franziska, ``Model-based inverse reinforcement learning from visual demonstrations," Conference on Robot Learning, pp. 1930-1942, 2021.

---

### Official Review · Reviewer_cHBi · 2022-07-11

**Rating:** 5
**Confidence:** 3
**Soundness:** 3 good
**Presentation:** 2 fair
**Contribution:** 3 good

**Summary:**

This paper considers a distributed inverse constrained RL problem formulated as a distributed bi-level optimization problem. It proposes a bi-level distributed algorithm with theoretical guarantees.

**Questions:**

1. Is it true that every learner knows global action and state spaces?  If it is the case, is there any application examples for such a setting?

2. In distributed RL algorithms, if the underlying matrices are doubly stochastic and assuming global action and state spaces, the distributed algorithm can be simplifies to a single agent counterpart by looking at the average of all agents' states. Is this also the case in this distributed inverse RL algorithm?

3. It is stated that Lines 5 - 10 in the algorithm are to track the average global gradient. Is it an inner loop averaging process? does the average need to be accurately achieved?

**Strengths And Weaknesses:**

The paper considers a very interesting problem in RL. The algorithm is novel.

The notation of the paper quite heavy and thus makes it not easy to follow.

My major concern is the technical novelty of the paper. The paper assumes doubly stochastic matrices in both inner and outer loops, and global observability of state and action spaces, which seem significantly simplifies the distributed algorithm design and analysis.

---

> ### Author Response · Authors · 2022-08-02
> **Response to Reviewer cHBi**
>
> Thank you for your constructive feedback. We believe that this discussion will help improve our paper overall. Before addressing the comments, we would like to first point out two significant differences between distributed reinforcement learning (RL) [B1] [B2] [B3] and distributed inverse reinforcement learning (IRL).
>
> Firstly, distributed RL and distributed IRL study different entities and problems. Distributed RL aims to design a distributed algorithm for a group of agents to learn a policy which can maximize the global (average) cumulative reward. The agents in distributed RL correspond to the experts in distributed IRL. In distributed IRL, the learners observe the behaviors of all the experts and use some algorithm to imitate the experts from the demonstrations. The learners do not need to know how the policy used by the experts is generated. In contrast, distributed RL focuses on distributed synthesis of such policy. In a nutshell, distributed RL focuses on the experts/agents and distributed IRL focuses on the learners. Moreover, (centralized) IRL usually assumes that the learner can get access to an RL oracle (either centralized or distributed) to solve the decision making problem [B4] [B5], so that how the policy is generated by RL algorithms is not important in IRL.
>
> Secondly, the data distributions are also different. In distributed RL, the data is distributed among the agents where each agent can only get access to local information including local reward and local action and global information including global state space [B2]. In distributed IRL, the data is distributed among the learners such that each learner only knows a local subset of the whole demonstration set but each local demonstration consists of the global information including a sequence of global state-action pairs. In conclusion, the global information in distributed RL refers to the information of the whole Markov decision process or Markov game. The global data in distributed IRL refers to the whole demonstration data set and each data in a local data set contains the global information in the sense of distributed RL.
>
> We use the following example to illustrate the differences between distributed RL and distributed IRL. Consider the patroller-evader scenario in [B6] where there are two patrollers patrolling around an area and there is an evader hidden outside the patrolled area who wants to learn the patrolling strategy and evade the area without being detected. The two patrollers are the experts and their patrolling policy can be obtained via solving some RL problem. If the patrollers aim to solve the RL problem in a distributed way, then it becomes a distributed RL problem. The evader is the learner and observes the states and actions of both patrollers, i.e., global states and actions, and how it can imitate the patrolling policy is the problem of IRL. If there are multiple evaders, it becomes a distributed IRL problem.
>
> Now, we address your comments below:
>
> $Weakness$: My major concern is the technical novelty of the paper. The paper assumes doubly stochastic matrices in both inner and outer loops, and global observability of state and action spaces, which seem significantly simplifies the distributed algorithm design and analysis.
>
> $Answer$: Thank you for mentioning the concern for the technique novelty. We agree that global observability of the state and action spaces will significantly simplify the distributed RL problem, however, this is not the case for distributed IRL. In distributed IRL, we only assume that the learners know the global state and action spaces. This is a standard assumption in multi-agent IRL literature [B6] [B7] [B8] [B9]. In the above patroller-evader example, this assumption is reasonable. The experts/agents can still execute existing distributed RL algorithms which only requires local action space. We also notice that the additional double-stochasticity on adjacent matrices will also simplify the distributed RL problem in the sense that the distributed agents can be reduced to a single agent, however, this is not the case for distributed IRL. In distributed IRL, the local demonstration and local estimate of the reward feature expectation and cost expectation cannot be shared (lines 147-148) and each local data already contains the information of all the experts. Our distributed ICRL algorithm will reduce to a single learner only when each learner knows information of the global demonstration set, e.g., either global demonstration data or global estimates of reward feature expectation and cost expectation. However, this is apparently not our case.
>
> The extension of relaxing the assumptions on global observability and doubly-stochastic matrices is interesting and worth exploring. However, even with these two assumptions, the paper requires more than 12 pages of proofs in the appendix. It clearly shows that the current setup is challenging enough.

---

> > ### Author Response · Authors · 2022-08-02
> > **Response to Reviewer cHBi (continued)**
> >
> > $Q1$: Is it true that every learner knows global action and state spaces? If it is the case, is there any application examples for such a setting?
> >
> > $Answer$: Yes, you are right. The learners know global state and action spaces, which is standard in multi-agent IRL literature [B6] [B7] [B8] [B9]. There are multiple examples supporting this. The first example is the patroller-evader setting [B6] discussed above. The second example is the traffic control example in [B7] where the (implicit) learner observes the states (cars density) and actions (traffic signal) of all the four agents (traffic lights at the intersections). The other examples in [B7] [B8] [B9] also fit this setting.
> >
> > $Q2$: In distributed RL algorithms, if the underlying matrices are doubly stochastic and assuming global action and state spaces, the distributed algorithm can be simplifies to a single agent counterpart by looking at the average of all agents' states. Is this also the case in this distributed inverse RL algorithm?
> >
> > $Answer$: Thank you for mentioning this. Please refer to our answer to the weakness.
> >
> > $Q3$: It is stated that Lines 5 - 10 in the algorithm are to track the average global gradient. Is it an inner loop averaging process? does the average need to be accurately achieved?
> >
> > $Answer$: Thank you for mentioning this averaging process. This is not an inner loop averaging process but an averaging process of the global gradient of the outer problem which needs the result obtained in the inner problem. The average does not need to be accurately achieved at each outer iteration but will approach the accurate average as the outer iteration number increases. In specific, in Algorithm 1, each $\Delta^{[v]}$ will converge to $\frac{1}{N_L}\sum_{v=1}^{N_L}\bar{\nabla}F^{[v]}(\bar{\omega}_c,\eta^{\ast}(\bar{\omega}_c))$ where $\bar{\omega}_c$ and $\eta^{\ast}(\bar{\omega}_c)$ are the consensus of outer and inner loop. The consensus achieving is guaranteed in Lemma 3 and Theorem 1.
> >
> > [B1] Kar, Soummya, Jose MF Moura, and H. Vincent Poor,``Qd-learning: A collaborative distributed strategy for multi-agent reinforcement learning through consensus," IEEE Transactions on Signal Processing, vol. 61, no. 7, pp. 1848-1862, 2013.
> >
> > [B2] Zhang, Kaiqing, Zhuoran Yang, Han Liu, Tong Zhang, and Tamer Basar, ``Fully decentralized multi-agent reinforcement learning with networked agents," International Conference on Machine Learning, pp. 5872-5881, 2018.
> >
> > [B3] Zhang, Kaiqing, Zhuoran Yang, and Tamer Basar, ``Multi-agent reinforcement learning: A selective overview of theories and algorithms," Handbook of Reinforcement Learning and Control, pp. 321-384, 2021.
> >
> > [B4] Abbeel, Pieter, and Andrew Y. Ng, ``Apprenticeship learning via inverse reinforcement learning," International conference on Machine Learning, pp. 1--8, 2004.
> >
> > [B5] Arora, Saurabh, and Prashant Doshi, ``A survey of inverse reinforcement learning: Challenges, methods and progress," Artificial Intelligence, vol. 297, p. 103500, 2021.
> >
> > [B6] Bogert, Kenneth and Doshi, Prashant, ``Multi-robot inverse reinforcement learning uner occlusion with interactions," International Conference on Autonomous Agents and Multi-agent Systems, pp. 173-180, 2014.
> >
> > [B7] Natarajan, Sriraam and Kunapuli, Gautam and Judah, Kshitij and Tadepalli, Prasad and Kersting, Kristian and Shavlik, Jude, ``Multi-agent inverse reinforcement learning," International Conference on Machine Learning and Applications, pp. 395-400, 2010.
> >
> > [B8] Reddy, Tummalapalli Sudhamsh and Gopikrishna, Vamsikrishna and Zaruba, Gergely and Huber, Manfred, `` Inverse reinforcement learning for decentralized non-cooperative multiagent systems," IEEE International Conference on Systems, Man, and Cybernetics, pp. 1930-1935, 2012.
> >
> > [B9] Yu, Lantao, Jiaming Song, and Stefano Ermon, ``Multi-agent adversarial inverse reinforcement learning," International Conference on Machine Learning, pp. 7194-7201, 2019.

---

> > > ### Comment · Reviewer_cHBi · 2022-08-09
> > > **Reply to author response**
> > >
> > > Thank the authors for answering my questions in detail.  I thus increase my score.

---

### Official Review · Reviewer_SMXn · 2022-07-16

**Rating:** 8
**Confidence:** 4
**Soundness:** 3 good
**Presentation:** 4 excellent
**Contribution:** 4 excellent

**Summary:**

The paper considers inverse RL (IRL), in a distributed setup with multiple experts demonstrating for multiple learners and includes constraints (areas of the state space that are not allowed).  The general model is setup, building on other IRL work, now generalizing to include multiple collaborative learners.  A multi-agent distributed learning algorithm is developed, based on a bi-level optimization approach.  The learning agents share features (they do not share their separate observations of the multiple experts).  The learners simultaneously estimate policies based on reward, and also constraints. The inner-outer loops run at different time scales (the outer loop is slower), and consensus iterations are used for collaboration among the learners.  An ‘order-of’ convergence proof is given under some reasonable assumptions.  Two experimental cases are studied in detail (a grid-world and a drone flying example), and in the drone case the experts are humans providing trajectories in gazebo. Several metrics are evaluated, and the new algorithm is compared with various benchmarks, such as a greedy method.

**Questions:**

How is the communications connectivity modeled or defined (how is the graph determined)?  (I think a local neighbor distance, but not sure I saw this explicitly.) This is time varying?  (Not clear in the experiments.) Any sense of how the communications topology affects the performance, convergence, etc.?

Not quite clear on C-IRCL: this has a centralized single processor (so no consensus), correct?  Whereas if you run D-IRCL with a fully connected network, then you would still use consensus round(s)?  Can you compare these?  And importantly, what is an optimal centralized solution (the most important benchmark)?

Not quite clear what is meant by "action constraint" (discussion of Fig  1).

Please clarify how Fig 1g is generated. Do you fix the number of consensus iterations and run the entire learning algorithm?  (And I think the relation of inner-outer time scales in the experiments wasn't clear, or I missed it.)

Grid-world: It is interesting that you picked a different number of expert trajectories for each learner (10,20,30,40).  All of these are independently generated?  What happens, for example, if all agents have the same number?  Or if an agent has very few (or even zero), then it will rely on consensus to learn and might slow convergence because of poor initial estimation, correct?

Grid-world: I don’t quite see a specific model for the expert policies, or their eventual knowledge of the environment.  From the Appendix, actually the experts are the same and acting independently unless they are close (and then some collision avoidance occurs)? The expert policy is stochastic, which accounts for CVR not being zero?

Also for the experts: Terminating a trajectory on collision (in grid world) is in effect an infinitely negative reward, or actually highly informative for finding the constraint regions?  Wouldn’t it be simpler to have some agents learn the constraints as an independent first step (it seems exactly like related earlier single agent RL work that is focused on “safety” metrics, e.g., Abeel)?  It seems collisions are to be thought of as absolute disaster, yet the experts do this with some finite probability.  (The reviewer does recognize that a goal here is to work from human experts, and it may be different in that context.)

6.2 drone example: “distribute the demonstration to two learners”.  How? Both learners have all of them?

Section 7: Future work -- The linear combination question is interesting.  Ultimately, is there more to learn with a nonlinear model?  I suppose this is strongly linked to the particular application setup?


**Limitations:**

No potential negative social impact is apparent.

**Strengths And Weaknesses:**

The paper breaks new ground by adding multiple collaborative learners to the IRL problem.  Overall, the paper has a nice balance of ideas, models, analysis, and experiments.  The work is nicely presented and very readable, with the models and algorithm well and clearly defined.  The notation is consistent with other works on IRL, which aids the reader.   The convergence proof is interesting and reveals aspects of asymptotic convergence rate. The experiments are well done and comparisons are interesting.

An interesting aspect is how the learners infer the constraints, even if an expert has never visited that state.

There do not appear to be any significant issues with this work.  Scalability, communications among the agents,  and the distributed method of optimization and convergence rate are all aspects that will likely be interesting to explore going forward.

The questions below may be useful to add clarity and interpretation, especially for the experiments.

---

> ### Author Response · Authors · 2022-08-02
> **Response to Reviewer SMXn**
>
> Thank you for your encouraging and constructive review, which provides an excellent summary of our work and promising directions for further research. We appreciate that you find this work novel and impactful and we believe that this discussion will contribute to a better paper in general. We address your comments below:
>
> $Q1$: How is the communications connectivity modeled or defined (how is the graph determined)? (I think a local neighbor distance, but not sure I saw this explicitly.) This is time varying? (Not clear in the experiments.) Any sense of how the communications topology affects the performance, convergence, etc.?
>
> $Answer$: This communication topology models two classes of networks. The first one is networks of static agents [A1] such as a large computer network where communication links between computers are predetermined. The communication topology could be time-varying as some communication links could be occasionally lost and then reestablished. The second one is networks of mobile agents [A2] like robots where an agent can only communicate with its close neighbors. The communication topology depends on the locations of mobile agents and is thus time-varying. This communication model has been widely used in distributed learning literature [A3][A4][A5].
>
> In the first experiment, we use a time-varying communication graph and we provide the corresponding time-varying adjacency matrix in Section 10 in the appendix. The adjacency matrix reflects the communication network where the entry at $i$-th row and $j$-th column of the adjacency matrix is nonzero if learners $i$ and $j$ can communicate. Detailed prescription of the relation between the communication graph and adjacency matrix is given in lines 103-112 in Section 2.
>
> The communication topology will not affect the asymptotic convergence in Theorem 1 as long as it satisfies the assumptions in lines 113-120. However, it can significantly affect the constants $C_1^{[v]}$ and $C_2^{[v]}$ associated with the convergence rate in Lemma 3. In the appendix (Subsection 9.4), we can see the full expression of the formula in Lemma 3 and find that the convergence of the inner loop will be slower if the number B in Assumption 1 increases and the number $N_L$ of the learners increases.
>
> $Q2$: Not quite clear on C-IRCL: this has a centralized single processor (so no consensus), correct? Whereas if you run D-IRCL with a fully connected network, then you would still use consensus round(s)? Can you compare these? And importantly, what is an optimal centralized solution (the most important benchmark)?
>
> $Answer$: Yes, you are correct. In C-ICRL, there is only a single central learner who obtains all the demonstration data, therefore, there is no communication and no consensus. A fully connected network is a special case of our time-varying network and we still need to use consensus rounds. It is interesting to compare D-ICRL with a fully connected network and C-ICRL, and we will do it in the future.
>
> The optimal centralized solution is C-ICRL and the pseudo code of C-ICRL can be obtained from the pseudo code of D-ICRL by the following reductions: (1) remove lines 2-3, 5-10, and 12-15; (2) the inner process at line 4 (i.e., Algorithm 2) becomes a standard (K-1)-step gradient descent process with update law $\eta(\omega_c,k+1)=\eta(\omega_c,k)-\alpha(k)\nabla_{\eta}G(\eta(\omega_c,k);\omega_c)$; (3) $N_L\bar{\nabla}^{[v]}(n)$ at line 11 becomes $\bar{\nabla} F(\omega_c(n),\bar{\eta}(\omega_c(n)))$.
>
> $Q3$: Not quite clear what is meant by "action constraint" (discussion of Fig 1).
>
> $Answer$: The action constraint, following the one in [A6], is the action that cannot be taken. For example, in the ground truth MG (Figure 1 (a)), the actions of moving diagonally to the bottom left and bottom right directions (indicated by the red crosses) of expert 1 is forbidden which means that expert 1 cannot move diagonally to the bottom left and bottom right directions. Our learners can find that expert 1 cannot move diagonally to the bottom right direction (shown in Figure 1 (b)).
>
> $Q4$: Please clarify how Fig 1g is generated. Do you fix the number of consensus iterations and run the entire learning algorithm? (And I think the relation of inner-outer time scales in the experiments wasn't clear, or I missed it.
>
> $Answer$: In our bi-level learning algorithm, the inner iteration is much faster than the outer iteration (line 112). In experiment 1, we run 50 outer iterations where each outer iteration requires 120 inner iterations. Figure 1 (g) is generated on the 120 inner iterations at the 50-th (last) outer iteration (lines 285-286) because at this time the consensus in the outer iteration is reached.

---

> > ### Author Response · Authors · 2022-08-02
> > **Response to Reviewer SMXn (continued)**
> >
> > $Q5$: Grid-world: It is interesting that you picked a different number of expert trajectories for each learner (10,20,30,40). All of these are independently generated? What happens, for example, if all agents have the same number? Or if an agent has very few (or even zero), then it will rely on consensus to learn and might slow convergence because of poor initial estimation, correct?
> >
> > $Answer$: Yes, your understandings are exactly on that point. All the trajectories are generated independently. When the learners have the same number of trajectories, they are more likely to contribute equally to the learning process. Consider the extreme case that one learner has zero data, then from the distributed algorithm (line 9 in Algorithm 1 and line 4 in Algorithm 2), we can see that this learner basically has no contribution to other learners because its local update gradient is always $0$ and thus will drag down the whole learning process.
> >
> > $Q6$: Grid-world: I don’t quite see a specific model for the expert policies, or their eventual knowledge of the environment. From the Appendix, actually the experts are the same and acting independently unless they are close (and then some collision avoidance occurs)? The expert policy is stochastic, which accounts for CVR not being zero?
> >
> > $Answer$: Thank you for mentioning the experts' model. There are two types of experts in experiment $1$ for comparison. The experts (ground truth) know everything (e.g., reward, constraint, etc) in the ground truth MG and the experts (nominal) do not know the constraints (lines 246-247). The experts are programmed to follow the optimal policy (lines 234-235) calculated by Q-learning under the environment they know.
> >
> > Yes, you are right. The experts act independently unless they are close since they want to avoid collision. Because the constraints are hard, the experts (ground truth) have 0 CVR even if they may have multiple optimal actions on some states and choose an optimal action randomly. The experts (nominal) have high CVR because they do not know any constraint.
> >
> > $Q7$: Also for the experts: Terminating a trajectory on collision (in grid world) is in effect an infinitely negative reward, or actually highly informative for finding the constraint regions? Wouldn’t it be simpler to have some agents learn the constraints as an independent first step (it seems exactly like related earlier single agent RL work that is focused on “safety” metrics, e.g., Abeel)? It seems collisions are to be thought of as absolute disaster, yet the experts do this with some finite probability. (The reviewer does recognize that a goal here is to work from human experts, and it may be different in that context.)
> >
> > $Answer$: Thank you for mentioning this. The environment will return a very large cost and terminate the episode if a collision happens. It is very interesting to first learn the constraints to simplify the RL problem. Since our focus is IRL, we use a straightforward revised Q-learning algorithm to find the policy for the experts. The experts (ground truth) have zero probability for collision (shown in Table 1) and the experts (nominal) have high probability for collision because they do not know any constraint.
> >
> > $Q8$: 6.2 drone example: “distribute the demonstration to two learners”. How? Both learners have all of them?
> >
> > $Answer$: We are sorry for our negligence as we accidentally removed the description for this part due to the page limit. In this experiment, there are $9$ pairs of demonstrations and the two learners get $4$ and $5$ pairs of demonstrations respectively. We include this information in the paper now (highlighted in blue).
> >
> > $Q9$: Section 7: Future work -- The linear combination question is interesting. Ultimately, is there more to learn with a nonlinear model? I suppose this is strongly linked to the particular application setup?
> >
> > $Answer$: Yes, we agree with you. Nonlinear models, e.g., neural networks, are more powerful and more general to model complicated functions but require more training data. Generally speaking, in applications where it is hard to find feature vectors for the latent function, a non-linear model can be a better choice than a linear one given that the data is abundant.

---

> > > ### Author Response · Authors · 2022-08-02
> > > **Response to Reviewer SMXn (continued)**
> > >
> > > [A1] Bertsekas, Dimitri and Tsitsiklis, John, ``Parallel and distributed computation: numerical methods," Athena Scientific, 2015.
> > >
> > > [A2] Cortes, Jorge and Martinez, Sonia and Karatas, Timur and Bullo, Francesco, ``Coverage control for mobile sensing networks," IEEE Transactions on Robotics and Automation, vol. 20, no. 2, pp. 243-255, 2004.
> > >
> > > [A3] Zhang, Kaiqing, Zhuoran Yang, Han Liu, Tong Zhang, and Tamer Basar, ``Fully decentralized multi-agent reinforcement learning with networked agents," International Conference on Machine Learning, pp. 5872-5881, 2018.
> > >
> > > [A4] Luo, Ping and Xiong, Hui and Lu, Kevin and Shi, Zhongzhi, ``Distributed classification in peer-to-peer networks," ACM SIGKDD International Conference on Knowledge Discovery and Data Mining, pp. 968-976, 2007.
> > >
> > > [A5] Lalitha, Anusha and Kilinc, Osman Cihan and Javidi, Tara and Koushanfar, Farinaz, ``Peer-to-peer federated learning on graphs," arXiv preprint arXiv:1901.11173, 2019.
> > >
> > > [A6] Scobee, Decter RR, and S. Shankar Sastry. ``Maximum likelihood constraint inference for inverse reinforcement learning." International Conference on Learning Representations. 2019

---

> > > > ### Comment · Reviewer_SMXn · 2022-08-09
> > > > **Thank you for your response**
> > > >
> > > > I appreciate the detailed comments, and the references are also very helpful.

---

> > ### Comment · Reviewer_SMXn · 2022-08-09
> > **Note on distributed connectivity**
> >
> > Thanks for your detailed responses.
> >
> > A general remark, that is unrelated to your specific work and not a criticism of your paper: We (as a multi-agent learning community) rely on consensus a great deal, which in effect transforms the problem into a centralized one.  Consensus is a very useful theoretical construct, because it gives an analytical guarantee under mild conditions.  However, the practical communications cost of consensus is far too high unless communications are essentially free (as in the wired network case).

---

### Meta-Review · Area_Chair_moy6 · 2022-08-26

**Recommendation:** Accept
**Confidence:** Less certain

**Metareview:**

The paper produces one of the first analysis of distributed inverse reinforcement learning, which is formulated as a bilevel optimization problem. The paper initiates a new line of research, contains a good mix of theoretical and empirical results, and has received relatively high scores from reviewers.

**Award:**

No

---

### Decision · Program_Chairs · 2022-09-14

Accept